# Lunar impact crater identification and age estimation with Chang'E data by deep and transfer learning

Chen Yang [1,2✉], Haishi Zhao [3], Lorenzo Bruzzone[4], Jon Atli Benediktsson [5], Yanchun Liang[3], Bin Liu [2], Xingguo Zeng [2], Renchu Guan [3✉], Chunlai Li [2✉] & Ziyuan Ouyang[1,2]

Impact craters, which can be considered the lunar equivalent of fossils, are the most dominant lunar surface features and record the history of the Solar System. We address the problem of automatic crater detection and age estimation. From initially small numbers of recognized craters and dated craters, i.e., 7895 and 1411, respectively, we progressively identify new craters and estimate their ages with Chang'E data and stratigraphic information by transfer learning using deep neural networks. This results in the identification of 109,956 new craters, which is more than a dozen times greater than the initial number of recognized craters. The formation systems of 18,996 newly detected craters larger than 8 km are estimated. Here, a new lunar crater database for the mid- and low-latitude regions of the Moon is derived and distributed to the planetary community together with the related data analysis.

[1] College of Earth Sciences, Jilin University, 130061 Changchun, China. [2] Key Laboratory of Lunar and Deep Space Exploration, National Astronomical Observatories, Chinese Academy of Sciences, 100101 Beijing, China. [3] Key Laboratory of Symbol Computation and Knowledge Engineering of Ministry of Education, College of Computer Science and Technology, Jilin University, 130012 Changchun, China. [4] Department of Information Engineering and Computer Science, University of Trento, I-38122 Trento, Italy. [5] Electrical and Computer Engineering, University of Iceland, 101 Reykjavik, Iceland. ✉email: yangc616@jlu.edu.cn; guanrenchu@jlu.edu.cn; licl@nao.cas.cn

The Moon's surface contains numerous impact craters that occupy most of the Moon's surface. Impact craters on the Moon span five lunar geologic time periods, i.e., the pre-Nectarian System, the Nectarian System, the Imbrian System, the Eratosthenian System and the Copernican System, spanning approximately four billion years. Their formation and evolution record the history of the inner Solar System[1–5]. Sixty years of advances in lunar exploration projects (e.g., the Luna missions and NASA's Apollo programme) have accumulated various lunar data, including digital images, digital elevation models (DEM) and lunar samples. Visual inspection of images and/or DEM data by experts or automatic detection[6–8] has recognized a large number of lunar craters, and consequently, many crater databases[9–13] have been established. However, the subjectivity of manual detection and the limitations of automatic detection with different types of data have resulted in significant disagreement in crater number among existing databases[14,15]. The International Astronomical Union (IAU)[16] has recognized 9137 lunar impact craters since 1919. The formation ages of 1675 of these lunar craters were aggregated by the Lunar and Planetary Institute (LPI)[17] in 2015, according to a professional paper, i.e., the Geologic History of the Moon (GHM)[18] from the United States Geological Survey (USGS), updated with the stratigraphy of craters database[19] (see Fig. 1a); at present, this represents a comprehensive database of craters ages.

The typical characteristics of a crater include large extents, differences in diameter on the scale of orders of magnitude, large variations in shape due to overlapping or filling, and variable and complex morphologies. Existing automatic detection algorithms[6–8] based on pattern recognition and machine learning (ML) can determine the large extent characteristic of craters from the general features of craters. Deep learning (DL), in particular convolutional neural networks (CNNs) applied for the extraction of fine-grained information, has been used for the identification of lunar craters[20,21]. DL has demonstrated fast and accurate performances based on huge amounts of detected craters as labelled samples. Nevertheless, available samples mainly include simple craters and thus do not represent irregular and seriously degraded craters that may have formed in early periods and provide an important historical record.

In lunar impact chronology, the relative age of lunar geologic units was first determined by the stratigraphic coverage relationship[22], the morphologic features of craters[23,24] and the optical maturity (OMAT)[25]. The stratigraphic coverage relationship is a basic and reliable method. In the absence of stratigraphic control, relative age is estimated according to the degradation and freshness of crater morphology, e.g., ray brightness, rim and terrace sharpness, rim texture, crater shape, crater walls and superimposed craters[26]. OMAT is based on the degree of modification of lunar soil by exposure to space and is applied to large rayed craters[27]. The absolute age was obtained by analyzing the crater cumulative size-frequency distributions (CSFDs)[28–30] and the radiometric ages of returned samples. These analyses have made OMAT a fundamental technique applicable to large craters. Recently, a crater age-rock abundance regression function was derived to estimate the ages of young craters (age ≤1 Ga) with diameters ($D$) ≥10 km by analyzing the thermophysical characteristics of lunar impact ejecta[31]. However, crater statistics with scarce lunar samples and complex impact events result in an extraordinarily difficult age estimation task. It is difficult to obtain satisfactory crater identification and age estimation results with a single type of data using conventional methods.

In the new generation of exploration of the Moon, the Chang'E-1 (CE-1) and Chang'E-2 (CE-2) orbiters[32,33] of China's Lunar Exploration Program (CLEP) have provided rich lunar data, acquiring two image sets with different spatial resolutions, i.e., 120 and 7 m digital orthophoto images (DOM), and 500 and 7 m DEM data. Obviously, large-scale data with a low resolution can capture the morphology of large craters, while high-resolution data are important for capturing the morphology of small craters. Transfer learning (TL)[34], one of the frontiers of ML that is motivated by the fact that humans can apply previous knowledge to solve new problems intelligently[35], has been successfully applied to problems in which sufficient training samples are not available[36–38]. Here, we convert the crater identification problem into a target detection task and progressively identify craters using the global lunar terrain products of CE-1 and CE-2 DOM and DEM by means of TL[34] using deep neural networks. Then, the estimation of crater age is mapped into a taxonomic structure, and the relative specific chronology of craters, i.e., the formation systems, is determined by combining crater morphological markers with stratigraphic information by an ensemble TL strategy. In the process of crater identification and age estimation, only the recognized craters of IAU and dated craters aggregated by the LPI are used for the training set for TL to ensure the generalization of the model.

## Results

**Multiscale lunar impact crater identification.** To comprehensively identify lunar impact craters, we proposed a two-stage crater detection approach with CE-1 and CE-2 data. The process of identification of lunar craters is described in Fig. 2. Considering the magnitude difference of crater scale, a series of crater images are taken from the CE-1 and CE-2 data corresponding to different spatial resolutions and sizes from different angles from −65° to 65° in latitude and from −180° to 65°, and 65° to 180° in longitude on the Moon (the mid- and low-latitude regions). These crater images are obtained by the fusion of DOM and DEM data (see Methods section for data consolidation). The DOM data present the morphological characteristics of craters, whereas the DEM data present topographic information. Three scales of images, i.e., 120 m CE-1 images with 5000 × 5000 and 1000 × 1000 pixels and 50 m CE-2 images with 1000 × 1000 pixels, are used to detect craters with large, medium and small diameter ranges, i.e., 50–600, 20–120 and 1–50 km (Fig. 2a). Adjacent images have a 50% overlap, and each crater may appear in two or three images.

In the stage of the detection approach (Fig. 2b), recognized craters in CE-1 images are randomly divided into three separate datasets, i.e., 5682, 1422 and 791 images for training, validation and testing, respectively, and all the recognized craters in CE-2 images, i.e., 6511 craters, are used for testing the second stage of the detection model. In the first stage, a partial network of the existing CNNs pre-trained with ImageNet data is transferred and reuses CE-1 data, according to a network-based deep TL method[38]. The first stage that uses CE-1 images with 5000 × 5000 pixels and 1000 × 1000 pixels achieves 94.71% recall, recovering almost all the recognized craters in the test set. Figure 2c shows the detection maps with CE-1 images (1000 × 1000 pixels). There are six adjacent maps that have 50% overlap with each other. Red squares represent the newly detected craters, and the red dashed squares represent the undetected craters that are on the image edge or are not fully displayed. However, these undetected craters can be detected in other adjacent maps. Then, in the second stage, we assume that no training data are available from CE-2, and the first-stage detection model is transferred to CE-2 images without any training sample. The learning procedure of the second stage follows transductive TL[35], which can derive learned features and knowledge for CE-2 data, obtaining a 93.35% recall. Finally, 117,240 craters that range in

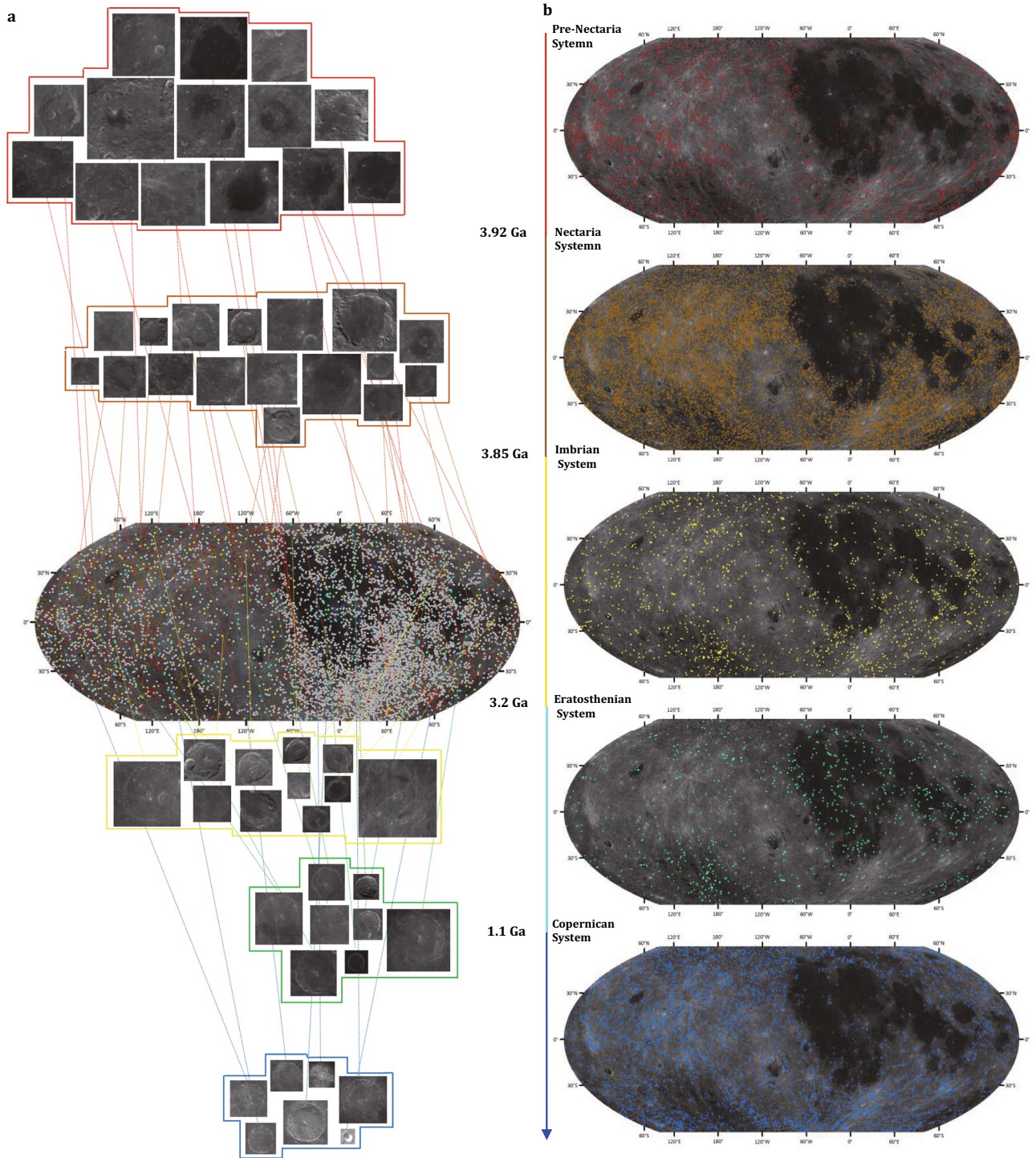

**Fig. 1 Distribution of lunar impact craters on the Moon. a** The distribution of recognized and dated craters. The red, brown, yellow, green and blue squares and points represent the craters of the pre-Nectarian System, the Nectarian System, the Imbrian System, the Eratosthenian System and the Copernican System, respectively. The grey points show recognized craters without ages. **b** Distribution of identified craters with assigned ages. From the time scale and spatial distribution, these dated craters exhibit specific characteristics. Craters with diameters smaller than 8 km and larger than 550 km are not shown in the distribution map.

size from approximately 0.9 to 532 km are identified. These craters are almost 15 times more than the recognized craters, and 88.14% of them are less than 10 km in diameter. Craters that appear in both CE-1 and CE-2 were removed by selecting the diameter $D \geq 20$ km for CE-1 detections and $D < 20$ km for CE-2 detections. The average detection time required for each image is 0.17 s. The catalogues of training, validation and test craters with

the DOM and the DEM data and the models of crater identification are publicly available at https://github.com/ hszhaohs/DeepCraters. The identified lunar craters can be found at https://doi.org/10.6084/m9.figshare.12768539.v1.

**Identified crater distribution and reliability assessment.** Figure 3 shows the distribution of identified craters by the detection

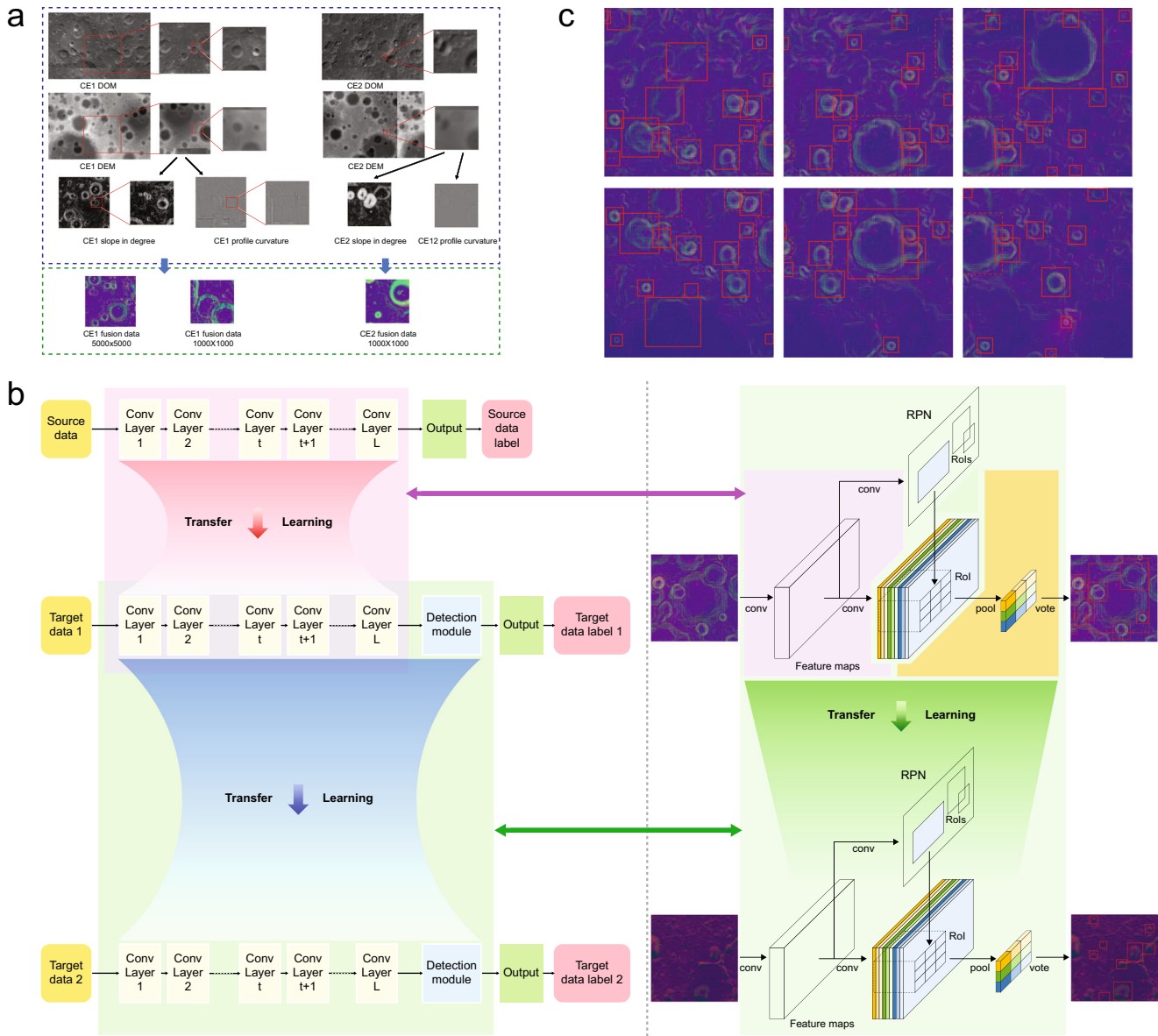

**Fig. 2 Identification of lunar impact craters based on transfer learning (TL) with Chang'E-1 (CE-1) data and Chang'E-2 (CE-2) data. a** CE-1 and CE-2 digital orthophoto images (DOM) and DOM data fusion and multiscale crater images. **b** The flowchart demonstrates the two-stage crater detection approach based on TL. The flowchart is split by a vertical dotted line. The left part shows the TL process, and the right part shows the corresponding network structure in detail. In the left part, the light pink area indicates the first-stage TL. The detection module is the region-based fully convolutional network with the basic network of the ResNet101 convolutional neural networks (CNN) architecture. ResNet101 (only the convolutional layers, not including the top fully connected layers) is transferred for crater detection. The detection module is fine-tuned by CE-1 data. Then, the detection module is directly transferred to CE-2 data, as shown in the light green area. It should be emphasized that there is no training in the second-stage TL. **c** Detection maps with CE-1 data. There are six adjacent detection maps that have a 50% overlap with each other. The red squares show the edge of detected craters, and the red dashed squares represent the individual undetected craters (on the image edge or not fully displayed on the image) in one of the detection maps. However, the individual undetected craters can be detected in the other adjacent detection maps.

model compared with the distribution of recognized craters. The number of identified lunar craters is systematically higher than that of recognized craters for diameters between 1 and 100 km. This indicates that the detection model finds a substantial number of craters with both small and medium diameter ranges. Although large craters can be irregular, seriously degraded and sparse on the Moon, 46 lunar craters with diameters ranging from 200 to 550 km were identified.

To verify the reliability of identified craters, we analysed identified craters in comparison with three manually derived lunar crater databases: (1) Head et al.[9] contains 5185 craters with

diameters $D \geq 20$ km obtained by using the digital terrain model (DTM) acquired by the Lunar Orbiter Laser Altimeter (LOLA) of the Lunar Reconnaissance Orbiter (LRO). (2) Povilaitis et al.[10] extended the above database to 22,746 craters with $D = 5$–20 km. (3) Robbins[11] has a database containing over 2 million craters with 1.3 million craters with $D \geq 1$ km and is the database with the largest number of lunar craters at present. Moreover, we considered three automated crater catalogues: (1) Salamunićcar et al.[12], i.e., LU78287GT, which was generated based on the Hough transform. (2) Wang et al.[13] compiled a global catalogue of 106,016 craters with $D > 500$ m using CE-1 data. (3) Silburt

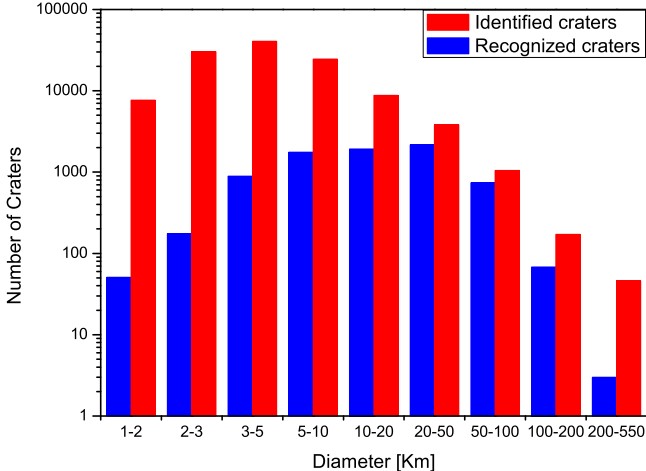

**Fig. 3 Identified lunar impact craters distributions compared with the recognized craters from the International Astronomical Union (IAU) in different diameter scales.** The red column represents the number of the identified craters compared with the number of the recognized craters (blue column). Recognized craters used for identification are the ones completely located within the study area having diameters larger than 1 km and smaller than 500 km. Source data are provided as a source data file.

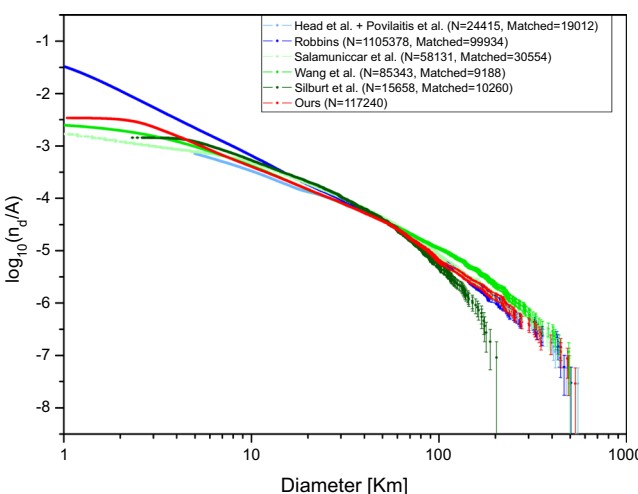

**Fig. 4 The cumulative size-frequency distributions (CSFDs) of craters in the existing databases and those identified in this paper.** The light blue and blue lines show the crater CSFDs of the manual lunar crater databases, i.e., combined Head et al. + Povilaitis et al. and Robbins. The light green, green and dark green lines show CSFDs of the three automated crater catalogues, i.e., Salamunićcar et al., Wang et al. and Silburt et al. The red line represents the CSFD of the craters identified in this paper. The confidence interval ±σ, which for the kth crater is $\log\left(\left(k \pm \sqrt{k}\right)/A\right)$, where A is the surface area, kth crater means that the kth-largest crater at the level of diameter. Source data are provided as a source data file.

et al.[21] generated a crater database using CNN with DEM data from the LRO. The comparison between craters detected in this paper and currently public lunar crater databases was carried out by matching criteria[11] in terms of both location and diameter: distance ≤ 0.25 × avg($D_i + D_j$), where $D_i$ and $D_j$ are the diameters of the ith and jth craters, respectively, and intersection over union IoU($C_i, C_j$) = ($C_i \cap C_j$)/($C_i \cup C_j$) ≥ 0.1, where $C_i$ and $C_j$ indicate the areas of the ith and jth craters, respectively. In the comparison process, two craters were recognized as matching when the two criteria were satisfied simultaneously. Figure 4 shows the CSFD of

the identified craters compared with the published lunar crater databases. The comparison results in terms of the number and percentage of matching craters at different scales are listed in Table 1.

For the manual databases, we can see that for databases of both Head et al.[9] and Povilaitis et al.[10], the CSFD and the matching numbers of identified craters show good agreement with our work. The matching percentage for both databases from 5 to 550 km is relatively stable. For the largest manual database, i.e., Robbins[11], it can be observed that the CSFD of craters between 1 and 20 km is systematically higher than that of the identified craters, whereas at larger diameters, the values are almost overlapping. It should be noted that the Robbins[11] database has many more craters than estimated in our work for D ≈ 1–20 km and a comparable number of craters for D ≈ 50–550 km. In terms of the matching percentage, most of the craters in this work are consistent with those in Robbins[11] when craters with D>3 km are considered, yielding a high agreement (85.24%). In general, our results achieve 85.30% agreement with published manual lunar crater databases for diameters D ≳ 1 km.

For the automated catalogues, the CSFD of Salamunićcar et al.[12] is lower than that of the identified craters when D ≲ 8 km. Our database has a significant mismatch for small craters D ≈ 1–3 km, whereas the agreement increases as diameter increases. This may be because small craters are not sufficient in that automated crater catalogue. For Wang et al.[13], the CSFD of craters between 1 and 5 km is lower than that of the identified craters, but at larger diameters, they are almost overlapping and are slightly higher when craters D > 100 km. However, the centre location of craters in Wang et al.[13] has different offsets from those of other databases due to the lack of global correction. Here, only the detected craters in CE-1 are used for comparison, representing ~10% of the total detected craters. The analysis shows that most of the craters extracted from CE-1 data have good consistency when D ≈ 10–50 km. For Silburt et al.[21], the curve origin at ~3 km increases gradually but cuts off sharply at diameters of ~200 km. The curve of the identified craters is relatively smooth and reaches ~532 km. This indicates that the TL-based detection approach finds a substantially larger number of craters (>7× more craters) than the only DL-based model in both small and large diameter ranges, which include faint, heavily degraded, and secondary craters that are often hard to detect with automated methods. Although there is a high matching assessment when D is between 50 and 550 km in overlapping areas, a disagreement appears when decreasing the diameter. This might be due to the crater detection in CNN, which is based on rectangular windows that cannot guarantee proper scaling to small crater diameters.

Then, the newly identified craters, i.e., 109,956 craters which are not included in the recognized craters, are divided into two sets of scales for manual assessment of detection accuracy. In the first set, all the newly identified craters with diameters larger than 100 km (i.e., 166) are involved in the assessment. In the second set, 10% of the other identified craters (i.e., 10,979) with diameters between 1 and 100 km are considered using a statistical sampling by random selection. These craters are assessed by matching with three manual databases and independently inspected by four scientists from Key Laboratory of Lunar and Deep Space Exploration, Chinese Academy of Sciences, simultaneously. The selected newly identified craters are projected onto CE-2 7m images and the mean ± standard deviation (s.d.) error results are provided. The false-positive rates (FPRs) of the newly identified craters are listed in Table 2. From Table 2, we can see that the FPRs of identified craters with D = 1–100 and D = 100–550 km are 4.49 ± 0.70% and 4.67 ± 2.10%, respectively. In the only DL-based model, i.e., Silburt et al.[21], the FPR of 361 new

**Table 1 Comparison between the identified craters and existing lunar crater databases in terms of number and percentage of matching craters for different diameter scales.**

|  | Head et al. + Povilaitis et al. | Robbins | Salamuniccar et al. | Wang et al. | Silburt et al. |
|---|---|---|---|---|---|
| 1–2 km |  |  |  |  |  |
| NA[a] | 0 | 7665 | 16,686 | 0 | 0 |
| NM[a] | 0 | 2592 | 4178 | 0 | 0 |
| MR(%)[b] | – | 33.82 | 25.04 | – | – |
| 2–3 km |  |  |  |  |  |
| NA | 0 | 30,332 | 7631 | 0 | 30 |
| NM | 0 | 23,498 | 4251 | 0 | 23 |
| MR(%) | – | 77.47 | 55.71 | – | 76.67 |
| 3–5 km |  |  |  |  |  |
| NA | 0 | 40,783 | 7150 | 258 | 2366 |
| NM | 0 | 38,402 | 5192 | 120 | 1744 |
| MR(%) | – | 94.16 | 72.62 | 46.51 | 73.31 |
| 5–10 km |  |  |  |  |  |
| NA | 13003 | 24,551 | 9719 | 2160 | 7458 |
| NM | 10210 | 22,418 | 6933 | 1370 | 5027 |
| MR(%) | 78.52 | 91.31 | 71.33 | 63.43 | 67.40 |
| 10–20 km |  |  |  |  |  |
| NA | 6859 | 8810 | 10,422 | 5666 | 3534 |
| NM | 5251 | 8356 | 6045 | 4866 | 2079 |
| MR(%) | 76.56 | 94.85 | 58.00 | 85.88 | 58.83 |
| 20–50 km |  |  |  |  |  |
| NA | 3246 | 3829 | 5023 | 2985 | 1841 |
| NM | 2522 | 3582 | 2881 | 2537 | 1004 |
| MR(%) | 77.70 | 93.55 | 57.36 | 84.99 | 54.54 |
| 50–100 km |  |  |  |  |  |
| NA | 1041 | 1052 | 1223 | 522 | 372 |
| NM | 827 | 890 | 877 | 257 | 327 |
| MR(%) | 79.44 | 84.60 | 71.71 | 49.23 | 87.90 |
| 100–200 km |  |  |  |  |  |
| NA | 223 | 172 | 230 | 55 | 56 |
| NM | 169 | 154 | 163 | 29 | 55 |
| MR(%) | 75.78 | 89.53 | 70.87 | 52.73 | 98.21 |
| 200–550 km |  |  |  |  |  |
| NA | 43 | 46 | 47 | 14 | 1 |
| NM | 33 | 42 | 34 | 9 | 1 |
| MR(%) | 76.74 | 91.30 | 72.34 | 64.29 | 100.00 |
| OMR(%)[b] | 77.78 | 85.24 | 52.56 | 78.80 | 65.53 |

All compared datasets are sampled from −65° to 65° in latitude and −180° to 180° in longitude on the Moon, and crater diameter is limited in the range of 1–550 km.
[a]NA is the number of craters in the existing lunar crater databases used for comparisons at different crater scales; NM is the number of matching craters in our lunar craters database.
[b]MR is the matching percentage between the compared datasets computed as the ratio between NM and NA for different diameter scales; OMR is the overall matching percentage computed as the ratio between the total NM and the total NA from 1 to 550 km.

**Table 2 False-positive rates of newly identified craters.**

|  | Num. newly identified | Num. assessment | Num. false detection[a] | FPR[b] |
|---|---|---|---|---|
| 1–2 km | 7663 | 766 | 64/37/20/44 | 4.49% ± 0.70% |
| 2–3 km | 30,320 | 3032 | 187/119/104/133 |  |
| 3–5 km | 40,348 | 4035 | 153/121/103/117 |  |
| 5–10 km | 22,843 | 2284 | 147/138/125/135 |  |
| 10–20 km | 6860 | 686 | 40/38/34/35 |  |
| 20–50 km | 1426 | 143 | 15/13/12/13 |  |
| 50–100 km | 330 | 33 | 8/5/4/6 |  |
| 100–200 km | 124 | 124 | 9/5/9/3 | 4.67% ± 2.10% |
| 200–550 km | 42 | 42 | 1/1/3/0 |  |

All the newly identified craters with diameters larger than 100 km are involved in assessment; 10% of the other newly identified craters between 1 and 100 km are considered using statistical sampling by random selection.
[a]Num. false detection is the number of craters that are supposed to not be craters by matching with the manual databases and scientists.
[b]FPR is the false-positive rate (mean ± s.d.) of the newly identified craters with $D = 100$–550 and $D = 1$–100 km.

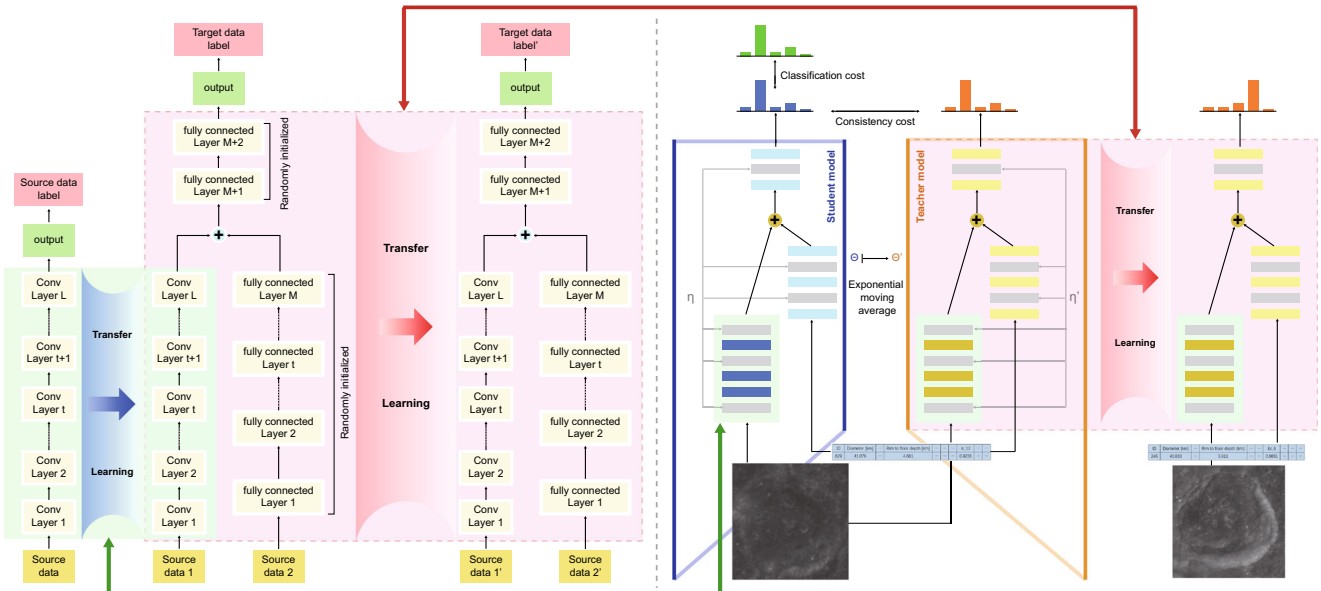

**Fig. 5 Estimation of the age of lunar craters based on transfer learning (TL) with Chang'E-1 (CE-1) data and Chang'E-2 (CE-2) data.** The flow chart demonstrates the two-stage crater classification approach based on TL. The two stages are separated by dotted lines. The left part is the TL process, and the right part shows the crater classification network structure. In the left part, the light green area indicates the first-stage TL. The crater classification model includes two types of input data, i.e., images and attribute data. Thus, the model consists of two channels. One of the channels is based on the pre-trained deep convolutional neural networks (CNN) model on ImageNet (only convolutional layers, not including the top fully connected layers) for images, and the other is the feedforward neural network for attribute data. Meanwhile, a semi-supervised learning strategy, i.e., Meanteacher, is adopted to take advantage of a large number of newly identified craters. In the second TL stage, the two-channel classification model with the Meanteacher strategy is directly used for estimating ages with CE-2 data, as shown in the light pink area. There is no other training in the second TL stage. A series of deep CNN techniques are then used for the classification of crater ages with CE images.

craters is 11 ± 7% derived also with manual inspection by four scientists and average results. This FPR latter is more than twice than that of the proposed model. Meanwhile, it should be noted that the number of detected craters of our TL-based model is obviously higher than that of DL-based model. This illustrates the reliability and stability of our detection model in the identification of lunar craters with both small and large sizes.

**Lunar impact crater age estimation.** The geological time scale established for the Moon was based on the recognition of convenient geomorphological markers with major impact events[18]. Absolute ages can be assigned to the geological periods by correlating the ages of samples obtained from Apollo missions. The pre-Nectarian System is defined as the period from when the lunar crust formed to when the Nectaris Basin was formed by a large impact, and its ejecta blanket serves as a useful stratigraphic marker. The Nectarian System and Imbrian System are defined by the occurrence of the Nectaris and Imbrium impact events, respectively. The Eratosthenian System is the period in which lunar craters can be recognized with freshly excavated materials on the lunar surface but with bright sputtering materials, i.e., rays around those craters, beginning to darken and disappear. The Copernican System is defined by craters generally surrounded by bright rays that represent recent lunar geologic records. It is important to note that some of the ages are uncertain, and we consider only the formation epoch. Here, the period boundaries of the five systems defined by Wilhelms[18] are used, i.e., the pre-Nectarian (>3.92 Ga), the Nectarian (3.92–3.85 Ga), the Imbrian (3.85–3.2 Ga), the Eratosthenian (3.2–1.1 Ga) and the Copernican (<1.1 Ga). Then, we mapped the five systems into a taxonomic structure.

As in the detection case, a two-stage crater classification approach based on TL with CE-1 and CE-2 data is proposed. The estimation of the age of the lunar crater scheme is shown in Fig. 5. A semi-supervised dual-channel lunar crater classification strategy is employed. One of the channels is used for extracting morphology marks with CNN; the other analyses the generic morphological information (e.g., diameter and depth) and stratigraphic attributes (i.e., coverage relationship) of craters (which cannot be directly derived from DOM data) with a feedforward neural network. The information on the scarce dated craters and enormous identified craters are simultaneously considered (see Methods section for age estimation algorithm).

In the first stage of the classification approach, 1411 dated craters with sizes ranging from approximately 1.26–1160 km in CE-1 were associated with the training, validation and test sets with proportions of 8:1:1. Twelve deep CNN models were transferred and fine-tuned with CE-1 data following the network-based deep TL technique[38]. For CE-2, the learning procedure of the second stage also followed transductive TL without any training data. A total of 502 craters from 1.26–50.66 km in the CE-2 images were used for testing the second stage of classification. The effectiveness of the classification approach in the first stage was tested with CE-1 data in five trials. The first stage of classification obtained an overall accuracy (OA) of 85.44 ± 1.94% (mean ± s.d.) (see Methods section for evaluation of ages of craters classification) and achieves the best OA of 88.97% on the test set of the dated craters. The corresponding confusion matrices of the CE-1 and CE-2 data are shown in Fig. 6. This demonstrates that the crater classification model trained in the first stage has the ability to accurately classify lunar craters into their respective systems. The best performance model in the first stage is transferred to CE-2 without training, resulting in 89.04% of dated craters being classified correctly. Finally, the best classification model is utilized to assign ages to identified and recognized craters larger than 8 km in diameter (because small craters degrade at an accelerated rate[23] with respect to large

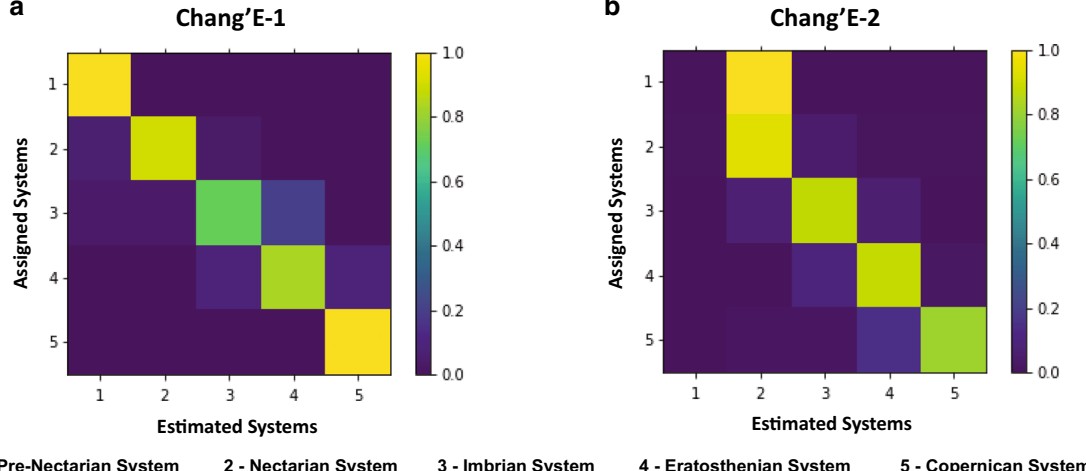

**1 - Pre-Nectarian System**  **2 - Nectarian System**  **3 - Imbrian System**  **4 - Eratosthenian System**  **5 - Copernican System**

**Fig. 6 Confusion matrix of the age estimation algorithm with Chang'E-1 (CE-1) data and Chang'E-2 (CE-2) data.** Confusion matrices for the crater age classification task of CE-1 and CE-2 data reveal acceptable misclassification of different systems. Element ($i$, $j$) of each confusion matrix represents the probability of estimating system $j$ given that the true system is $i$, with $i$ and $j$ referring to different systems. The diagonal of the matrix represents the probability of corrected classification for each system. Note that with both CE-1 and CE-2 images, there is some confusion between adjacent systems. For the first stage of classification with CE-1 images, compared with other systems, the pre-Nectarian System and the Copernican System have a very high accuracy (100%). Some of the craters of the Imbrian System are confused with those of the Eratosthenian System. For the second stage with the CE-2 image, only two craters of the pre-Nectarian System are used for testing owing to the resolution and confusion with those of the Nectarian System. The diameters of craters in the Eratosthenian System and the Copernican System are relatively small, and craters of the Copernican System sometimes do not feature bright rays in the DOM data. However, the overall classification results are accurate and reliable for supporting scientific analysis and interpretation. Source data are provided as a source data file.

craters of the same age). The average time of the classification of each crater is 0.006 s.

**Dated lunar impact crater distribution and mapping**. The dated lunar crater distribution is given in Fig. 7. Figure 7a compares the number of lunar craters with estimated ages to the recognized craters with formation systems. The results show that the lunar craters of the pre-Nectarian System are large in diameter (the number of craters with $D \geq 50$ km is far greater than those of other systems) in both our dataset and previous datasets. There are 18,996 craters in the study areas and ~1270 larger than 50 km. However, most of the recognized craters (733 out of 1453) in the pre-Nectarian System are distributed from 50 to 550 km. Similar results are obtained for the Nectarian System, which was associated with only 645 of the previous mapped craters. The number of dated craters in the Nectarian System is the highest, i.e., 11,050. However, there are more craters with diameters smaller than 20 km than those in the pre-Nectarian System. For the Imbrian System, the number of craters is 1431, and most of them are smaller than 50 km. For the two young geological periods, the number of craters in the Eratosthenian System, i.e., 850, and the Copernican System, i.e., 4212, are obviously different. Craters in the Eratosthenian System range in size from 20 to 50 km, whereas the craters in the Copernican System are mainly smaller than 20 km.

In Fig. 7b, we show the CSFDs of the craters with estimated ages by the TL-based classification model compared with those of recognized craters in the five systems. For the pre-Nectarian System, the CSFD curve of estimated craters first decreases slowly with increasing diameter and runs parallel to or overlaps the CSFD of recognized craters between 20 and 200 km. Then, it displays a prominent kink at 200 km and cuts off at diameters of ~500 km. The CSFD curve of estimated craters in the Nectarian System is clearly higher than that of the recognized craters for diameters $D \leq 50$ km, whereas the number of craters increases

slowly for diameters between 60 and 200 km, breaking at ~532 km (which is the largest identified crater with a radius of ~4433 pixels in the CE1 image). For diameter values between 8 and 30 km, the CSFD curves derived by the TL-based model for the Imbrian System are higher than the CSFD curves of recognized craters. Then, they overlap with each other for larger values and cut off at diameters of ~200 km. For the two relatively young systems, the TL-based model-derived CSFDs of the Eratosthenian System and the Copernican System are systematically higher than those of the recognized ones when the diameters of craters are relatively small ($D \leq 30$ km). Then, the newly dated CSFD curve follows the same size distribution as the recognized craters.

Two typical regions, i.e., nearside mare[10] and the northwest of the farside highlands[39], were selected for analyzing the superimposed crater populations associated with the five systems (Fig. 7c, d). Widespread basin resurfacing and basaltic flooding reset the vast majority of the nearside mare. The Imbrium basin formed at ~3.91 Ga[40] and provides an upper limit for the emplacement of basalts in this basin. This area has the lowest density of craters across the entire diameter range in existing manual databases[9–11]. In contrast, the northwestern farside highland region is one of the most heavily cratered terrains of the Moon for craters with diameters of both >20 (ref. [9]) and 5–20 km[10]. Meanwhile, this area is far from large impact basins (e.g., the South Pole-Aitken basin) and is unaffected by >5 km secondaries[39]. The "Relative" plots ($R$ plots) in Fig. 7c, d show a distinct difference between the shapes of the CSFDs of the ancient highlands and the young mare crater population. The farside highlands have a high density of large craters with $D \approx 50$–100 km in the pre-Nectarian System and the Nectarian System and of small craters with $D \approx 15$–20 km in the Copernican System. The density of the Imbrian System is intermediate, with craters with $D \approx 8$–50 km. A low density of craters associated with the Eratosthenian System ($D \approx 8$–20 km) is observed. For the nearside mare, the $R$ plots show a separable distinction between

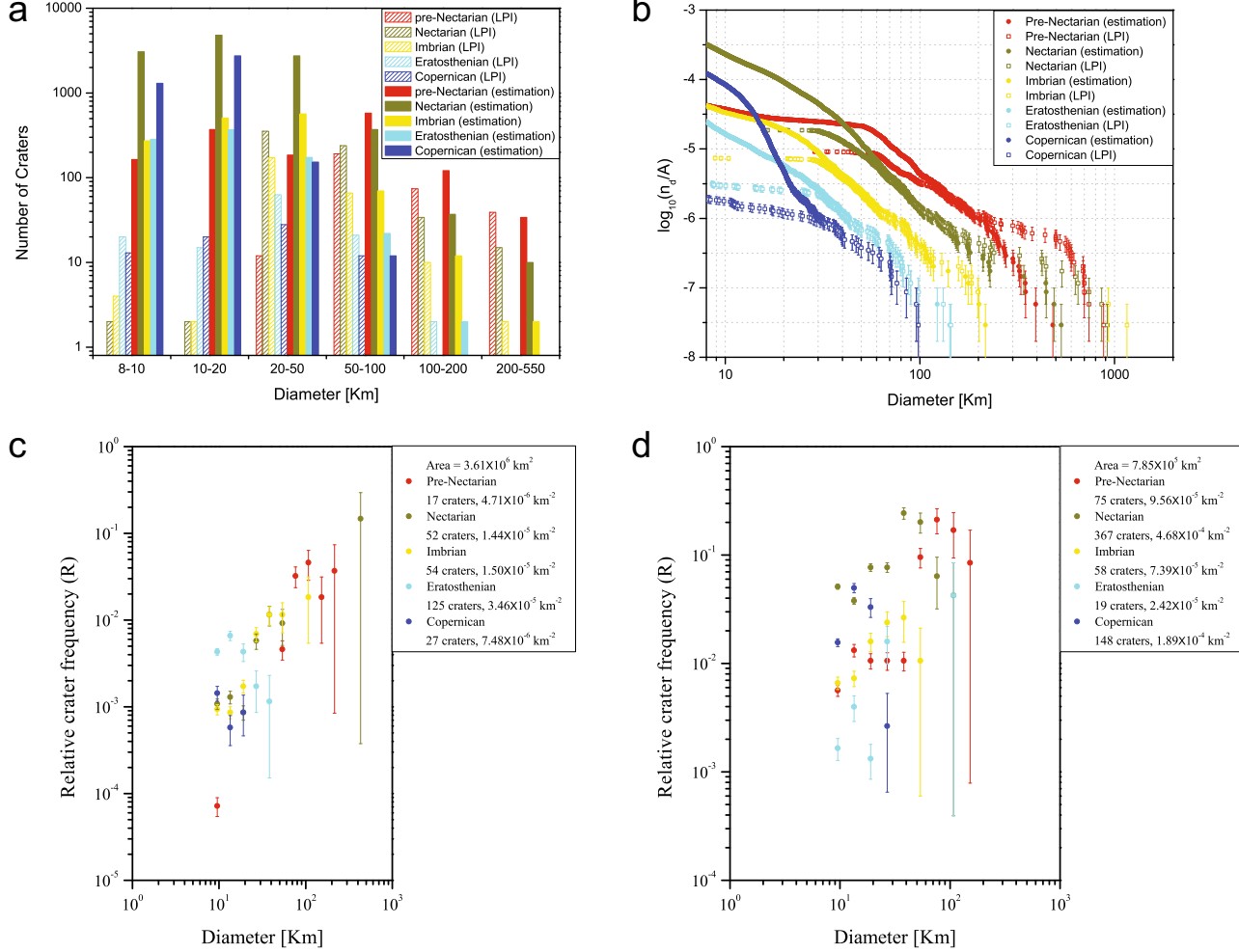

**Fig. 7 Dated lunar crater size-frequency distributions. a** Number of estimated craters and related assigned ages from the Lunar and Planetary Institute (LPI) with five ages in different diameter scales. Source data are provided as a source data file. **b** The cumulative size-frequency distributions (CSFDs) of identified craters with estimated ages and recognized craters with ages. Source data are provided as a source data file. The red, brown, yellow, green and blue lines represent the estimated crater CSFDs of the pre-Nectarian System, the Nectarian System, the Imbrian System, the Eratosthenian System and the Copernican System, respectively, and the hollow lines show recognized crater CSFDs of the five systems in the LPI used for age estimation. The confidence interval $\pm\sigma$, which for the $k$th crater is $\log\left(\left(k\pm\sqrt{k}\right)/A\right)$, where $A$ is the surface area, $k$th crater means that the $k$th-largest crater at the level of diameter. Source data are provided as a source data file. **c, d** $R$ plots (areal density) of dated craters superimposed on lunar nearside mare and farside highland areas, illustrating the difference in density and CSFD slope of the five systems on the two terrains. The confidence interval $\pm\sigma$ is $\log\left(R\pm R/\sqrt{N}\right)$, where $R$ is the $R$ value, $N$ is the cumulative number of craters. Source data for **c, d** are provided as a source data file.

the pre-Nectarian System and the Eratosthenian System. This area has a much higher crater density separable distinction between the pre-Nectarian System and the Eratosthenian System. This area has a much higher crater density of small craters with $D \approx 10$–$20$ km in the Eratosthenian System, whereas it shows the lowest density of large craters with $D \approx 100$–$200$ km in the pre-Nectarian System. The Nectarian System and the Imbrian System have similar crater frequencies. However, their $R$ curves are different in terms of the crater diameter range. This result indicates that the populations of craters in the pre-Nectarian System, the Nectarian System and the Imbrian System were significantly affected by the resurfacing events. The low density of the Copernican System is reflected in craters with $D \approx 8$–$20$ km. In general, the density of large craters ($D \approx 60$–$100$ km) is much higher than that of small craters ($D \approx 10$–$20$ km) in the farside highlands than in the nearside mare. This is consistent with a less-steep production function for this size range[9].

The spatial distribution of all the dated craters ($D \geq 8$ km) in the study area is shown in Fig. 1b. The spatial distribution varies

greatly for the five different systems. Craters of the pre-Nectarian System are widely distributed in the south and north and on the back of the Moon in the mid- and low-latitude regions. The craters in the Nectarian System are located on a larger area of the Moon except for lunar mare. Meanwhile, data from lunar samples (Apollo, Luna and lunar meteorites) indicate that the Moon was subjected to an intense period of bombardment, i.e., the Late Heavy Bombardment (LHB), at approximately 3.8–4.0 Ga[41,42].

Craters of the Imbrian System are mainly distributed on the front of the Moon and around lunar mare, mostly above the ejecta of mare and filled with mare basalts. This may be due to the hypothesis that many large-scale basaltic eruptions occurred after formation of the lunar mare (known as lunar mare flooding), which occurred between ~3 and 3.5 Ga[43]. The craters that formed during the Eratosthenian System are mainly distributed in the medium and high $TiO_2$ (ref. [44]) and $FeO$[45] basalt regions of the Mare Imbrium. From the chemical compositions of rocks and surface ages of mare basaltic units in Mare Imbrium estimated with CE-1 and Clementine ultraviolet to visible spectrum

(UVVIS) data, the evolution of the basalts is from low-titanium and low-iron compositions to high-titanium and high-iron compositions from the Imbrian System to the Eratosthenian System[46]. The impact in the Eratosthenian System may have been the external cause of the multistage volcanic eruption in the Imbrium basin. The results show that craters formed in the Copernican System are scattered all over the lunar surface in the study area. Recent research[47] has pointed out that sporadic meteoroid bombardment occurred across the whole Moon at ~800 Ma.

**Analysis of consistency with existing lunar age chronology.** The estimated ages for craters in this work are based on morphologic and stratigraphic information. Therefore, we compared dated craters with related literature using OMAT data[27], CSFDs[30] and thermophysical characteristics of lunar impact ejecta[31] described in the Introduction section.

Table 3 summarizes the comparison of relative crater ages derived from the OMAT data[27] and the assigned formation system ages based on the best classification model. The categories are listed on the basis of the shape of the OMAT profiles of crater

**Table 3 Comparison of crater relative ages derived from the optical maturity parameter (OMAT).**

| Crater name | Centre lat., lon. | Crater image | D, km | USGS epoch[a] | Category[b] | Estimated age |
|---|---|---|---|---|---|---|
| Green M | 0.4°N, 133.2°E | | 37.9 | Coper. | Old | Copernican |
| Briggs B | 28.3°N, 70.8°W | | 30.1 | Coper. | Old ? | Copernican |
| Stefan L+ | 44.4°N, 108.0°W | | 28.6 | Eratos. | Intermediate | Eratosthenian |
| Robinson | 59.1°N, 46.1°W | | 28.4 | Imbr. | Old | Nectarian |
| Horrebow | 58.8°N, 41.0°W | | 28.2 | Imbr./Eratos. | Old | Eratosthenian |
| Larmor Q | 24.8°N, 178.7°W | | 27.3 | Eratos. | Intermediate | Copernican |
| Gerasimovich D | 22.4°S, 122.0°W | | 27.1 | Imbr. | Old | Imbrian |
| Al-Khwarizmi K | 4.5°N, 108.2°E | | 26.8 | Coper. | Intermediate | Copernican |
| Lalandc | 4.4°S, 8.5°W | | 26.5 | Coper. | Intermediate | Copernican |
| Moore F | 37.3°N, 175.0°W | | 26.3 | Eratos. | Young | Copernican |
| Dufay B | 8.4°N, 171.2°E | | 26.2 | Coper. | Intermediate | Copernican |
| Focas | 33.7°S, 93.8°W | | 24.1 | Eratos. | Intermediate | Eratosthenian |
| 54S 161E | 53.9°S, 161.2°E | | 22.0 | Coper. | Intermediate | Copernican |
| Golitsyn J | 27.6°S, 103.2°W | | 21.9 | Eratos. | Old ? | Copernican |
| Byrgius A | 24.5°S, 63.7°W | | 21.9 | Coper. | Young | Copernican |
| 34S 130W | 34.2°S, 129.6°W | | 20.9 | Imbr. | Old | Eratosthenian |
| 43S 143E | 43.8°S, 143.4°E | | 19.1 | Nectar. | Intermediate | Copernican |

The compared craters are identified and recognized craters without assigned formation systems based on the best classification model. The position and diameters of crater centres were measured from the CE data.
[a]Stratigraphic epoch estimated in USGS Lunar Geologic Map Renovation (2013 edition).
[b]Old represents the craters older than Copernicus (inferred age of ~810 Myr), and Young is as young or younger than Tycho (inferred age of ~109 Myr), and Intermediate is between Old and Young in which Copernicus and Tycho are the oldest and youngest end-members, respectively.

**Table 4 Comparison with the crater age by the crater size–frequency distributions (CSFDs).**

| Crater name | Center lat., lon. | Crater image | $D$, km | USGS epoch[a] | LPI age | CSFDs age[b] | Estimated age | Fold[c] |
|---|---|---|---|---|---|---|---|---|
| Laue | 28°N, 97°W | | 88 | Imbr./Pr-Nectar. | pre-Nectarian | 3.9 ± 0.1 | pre-Nectarian | 4 |
| Freundlich | 25°N, 171°E | | 83 | Imbr./Nectar. | Nectarian | 4.0 ± 0.1 | Nectarian | 4 |
| Bridgman | 43°N, 137°E | | 82 | Imbr./Nectar. | Nectarian | 3.9 ± 0.1 | Nectarian | 1,2 |
| Langmuir+ | 36°S, 129°W | | 91 | Imbr. | Nectarian | 3.5 ± 0.1 | Nectarian | 3 |
| La Perouse | 11°S, 76°E | | 80 | Imbr. | Upper Imbrian | 3.6 ± 0.1 | Imbrian | 2 |
| Hahn | 31°N, 74°E | | 88 | Imbr. | Lower Imbrian | 3.8 ± 0.1 | Imbrian | 1 |
| Sinus Iridum | 43°S, 31°W | | 247 | Imbr. | — | ~3.8[36] | Imbrian | 2 |
| Theophilus | 12°S, 26°E | | 99 | Coper. | Eratosthenian | 3.0 ± 0.6 | Eratosthenian | 1 |
| Geminus | 34°N, 57°E | | 82 | Eratos. | Eratosthenian | 3.2 ± 0.4 | Eratosthenian | 2,5 |
| Aristoteles | 50°N, 17°E | | 88 | Eratos. | Eratosthenian | 2.7 ± 0.8 | Eratosthenian | 4 |
| Vavilov+ | 1°S, 139°W | | 98 | Coper. | Copernican | 1.7 ± 0.1 | Copernican | 5 |

The compared craters are recognized ones in the test set of the classification model. The position and diameters of crater centres were measured from the CE data.
[a]Stratigraphic epoch estimated in USGS Lunar Geologic Map Renovation (2013 edition).
[b]In Ga with 1r error.
[c]The classification model in five trials.

ejecta in which three relative age groups were classified, i.e., older than Copernicus (~810 Myr), intermediate and as young or younger than Tycho (~109 Myr). The estimated ages of craters are fitted well by the OMAT. We observe only one crater, i.e., Stefan L (marked with "+" in Table 3), with an estimated age that is not consistent with the category derived from OMAT. From Table 3, one can see that the Stefan L crater has a low ray brightness with a loss of rim texture. Additionally, the stratigraphic information estimated in the USGS maps is the Eratosthenian.

The craters in the test set of the classification model were selected for comparison of the absolute age by using CSFDs (Table 4). The previously assessed formation epochs obtained from the USGS Geologic Atlas of the Moon, the ages aggregated by the LPI, the absolute ages computed by CSFDs and the estimated age using the proposed TL-based classification model are given in the 4–7th columns. Table 4 is organized from old to young relative to the formation system (column 7). In total, the ages of 50% of the craters are estimated by the LPI and our method to be older than those in the USGS maps. The estimated ages of these craters are generally in the CSFD age range within error. We observe two craters, i.e., Langmuir and Vavilov (marked with "+" in Table 4), for which there are differences between the CSFDs and this work. From Table 4, one can see that the Langmuir crater features some small superimposed craters, and the rim and terrace sharpness show irregular degradation. The CSFD age of Langmuir may be affected by resurfacing; thus,

it is uncertain[30]. Vavilov shows clear geomorphic features with only a minimum of subsequent impact erosion but features weak rays, which may cause confusion with respect to the formation system.

Table 5 reports the comparison between craters with absolute ages by analysis of thermophysical characteristics of lunar impact ejecta[31] and those with estimated ages obtained by the best classification model. The absolute ages of all determined craters are located in the Copernican. A majority of compared craters (81%) that do not have any age information are also assigned to the Copernican. We observe 14 craters (marked with "+" in Table 5) with estimated ages that are not consistent with the absolute age. To obtain representative craters, four inconsistent craters, i.e., Euclides C, Sirsalis F, 16 and 31, and four consistent craters, i.e., Larmor Q, Cauchy, 15 and 22, were selected for comparison. The Euclides C and Sirsalis F craters are located in and around the lunar mare and show weak rays. Crater 16 features three small superimposed craters and a polygonal shape. In the bottom of crater 31, a sloping channel is observed on the crater wall. The comprehensive features extracted from the morphology and the stratigraphic information determined that these craters are hard to classify as Copernican. In contrast, Larmor Q, crater 15 and crater 22 are located in the highlands and have relatively intense rays that help decision-making. The rays of the Cauchy crater seem not evident for distribution in mare, but the comprehensive features suggest that it is Copernican.

**Table 5 Comparison of crater ages by analysis of thermophysical characteristics of lunar impact ejecta.**

| | Recognized lunar craters | | | | | No name lunar craters | | | |
|---|---|---|---|---|---|---|---|---|---|
| Crater name | Centre lat., lon. | D, km | Age[a] | Estimated age | Crater ID[b] | Centre lat., lon. | D, km | Age[a] | Estimated age |
| Copernicus (C) | 10°N, 20°W | 96 | 797 | Copernican | 1 | 15°N, 152°E | 21 | 449 | Copernican |
| Crookes | 10°S, 165°W | 48 | 446 | Copernican | 2 | 20°S, 117°E | 20 | 209 | Copernican |
| Proclus | 16°N, 47°E | 27 | 253 | Copernican | 3 | 22°S, 115°W | 20 | 221 | Copernican |
| Lalande | 4°S, 9°W | 27 | 495 | Copernican | 4 | 1°N, 159°W | 19 | 173 | Copernican |
| Moore F (MF) | 37°N, 175°W | 26 | 41 | Copernican | 5 | 49°N, 177°W | 18 | 377 | Copernican |
| Larmor Q | 29°N, 176°E | 26 | 178 | Copernican | 6 | 23°N, 80°E | 17 | 993 | Copernican |
| Cleostratus J[+] | 61°N, 84°W | 23 | 443 | Eratosthenian | 7 | 48°N, 154°W | 17 | 138 | Copernican |
| Innes G | 27°N, 122°E | 22 | 527 | Copernican | 8 | 44°N, 97°W | 16 | 1026 | Copernican |
| Byrgius A (BA) | 25°S, 64°W | 22 | 47 | Copernican | 9 | 15°N, 109°W | 16 | 113 | Copernican |
| Sundman V | 12°N, 94°W | 22 | 93 | Copernican | 10 | 26°N, 82°E | 15 | 406 | Copernican |
| Ventris M | 6°S, 158°E | 21 | 391 | Copernican | 11 | 63°S, 72°E | 15 | 539 | Copernican |
| Carrel[+] | 11°N, 27°E | 20 | 295 | Eratosthenian | 12 | 36°N, 166°E | 14 | 645 | Copernican |
| Mandel'shtam F | 5°N, 166°E | 19 | 44 | Copernican | 13 | 12°S, 150°W | 14 | 181 | Copernican |
| Ryder | 44°S, 143°E | 19 | 140 | Copernican | 14 | 29°S, 133°W | 14 | 81 | Copernican |
| Gauss J | 41°N, 73°E | 18 | 191 | Copernican | 15 | 14°N, 126°W | 14 | 34 | Copernican |
| Dawes | 17°N, 26°E | 18 | 454 | Copernican | 16[+] | 12°N, 168°W | 14 | 290 | Nectarian |
| Hume Z | 4°S, 90°E | 18 | 981 | Copernican | 17 | 6°N, 145°W | 13 | 678 | Copernican |
| Fechner T | 59°S, 123°E | 18 | 33 | Copernican | 18[+] | 45°S, 160°W | 13 | 708 | Eratosthenian |
| Geminus C | 34°N, 59°E | 17 | 800 | Copernican | 19 | 10°N, 170°W | 13 | 407 | Copernican |
| Darney C | 14°S, 26°W | 16 | 582 | Copernican | 20 | 1°N, 143°E | 13 | 318 | Copernican |
| Aitken A | 14°S, 173°E | 16 | 259 | Copernican | 21 | 48°S, 142°W | 13 | 563 | Copernican |
| Dugan J | 61°N, 108°E | 15 | 159 | Copernican | 22 | 12°N, 111°E | 13 | 238 | Copernican |
| Egede A | 52°S, 11°E | 15 | 84 | Copernican | 23 | 17°S, 94°E | 13 | 137 | Copernican |
| Cauchy | 10°N, 39°E | 15 | 489 | Copernican | 24 | 37°S, 132°E | 12 | 110 | Copernican |
| Papaleksi Q | 10°N, 163°E | 15 | 343 | Copernican | 25 | 11°N, 162°E | 12 | 385 | Copernican |
| Mohorovicic F | 19°S, 163°W | 15 | 385 | Copernican | 26[+] | 23°N, 35°E | 11 | 873 | Imbrian |
| Lagrange D | 35°S, 73°E | 15 | 257 | Copernican | 27 | 16°N, 177°E | 11 | 39 | Copernican |
| Pico B[+] | 47°N, 15°W | 14 | 612 | Eratosthenian | 28 | 44°N, 124°W | 10 | 369 | Copernican |
| O'Day M | 32°S, 157°E | 14 | 436 | Copernican | 29[+] | 21°N, 109°E | 10 | 226 | Nectarian |
| Eichstadt G | 22°S, 81°W | 14 | 686 | Copernican | 30 | 60°N, 174°E | 10 | 145 | Copernican |
| Lagrange H | 29°S, 66°W | 14 | 583 | Copernican | 31[+] | 35°S, 153°W | 9 | 587 | Imbrian |
| Alfraganus C | 6°S, 18°E | 13 | 433 | Copernican | | | | | |
| Euclides C[+] | 13°S, 30°W | 13 | 50 | Eratosthenian | | | | | |
| Avery[+] | 1°S, 81°E | 13 | 139 | Eratosthenian | | | | | |
| Aratus | 24°N, 5°E | 13 | 421 | Copernican | | | | | |
| Mercurius H | 49°N, 63°E | 13 | 181 | Copernican | | | | | |
| Nicolai A[+] | 42°S, 24°E | 12 | 669 | Nectarian | | | | | |
| Democritus A | 62°N, 32°E | 12 | 218 | Copernican | | | | | |
| Sirsalis F[+] | 14°S, 60°W | 12 | 629 | Nectarian | | | | | |
| Beaumont D | 17°S, 26°E | 12 | 511 | Copernican | | | | | |
| Fraunhofer G | 39°S, 58°E | 11 | 454 | Copernican | | | | | |
| Hilbert A[+] | 16°S, 109°E | 11 | 301 | Nectarian | | | | | |
| Reimarus H[+] | 49°S, 62°E | 10 | 640 | Nectarian | | | | | |

| Crater Name | Crater Image | D, km | USGS epoch[c] | Estimated age | Crater ID | Crater Image | D, km | USGS epoch[c] | Estimated age |
|---|---|---|---|---|---|---|---|---|---|
| Euclides C[+] |  5 km | 13 | Eratos. | Eratosthenian | 16[+] |  5 km | 14 | Nectar. | Nectarian |
| Sirsalis F[+] |  5 km | 12 | Imbr. | Nectarian | 31[+] |  5 km | 9 | Imbr. | Imbrian |
| Larmor Q |  8 km | 26 | Eratos. | Copernican | 15 |  5 km | 14 | Imbr. | Copernican |
| Cauchy |  5 km | 15 | Coper. | Copernican | 22 |  5 km | 13 | Nectar. | Copernican |

The compared craters are identified and recognized ones without assigned formation systems based on the best classification model. The position and diameters of crater centres were measured from the CE data.
[a]Ages in Myr estimated by analysis of thermophysical characteristics of lunar impact ejecta.
[b]No name craters with assigned number.
[c]Stratigraphic epoch estimated in USGS Lunar Geologic Map Renovation (2013 edition).

## Discussion

Through the proposed two-stage detection and classification approaches, an enormous number of craters were identified, and starting from the limited number of recognized craters, their ages were estimated. There are many reasons to argue that the TL-based strategy with a deep neural network has properly learned the complex features that define a crater and its age. For identified craters, despite the large variations in size and shape across the lunar surface, our approach was able to recover 94.71 and 93.35% of craters already known and that were not used to train our model previously. Moreover, it detected a number of new craters dozens of times larger than the number of recognized craters throughout the mid- and low-latitude regions of the Moon. Meanwhile, most of the craters known in the existing manual database and automated catalogues were also accurately detected by the proposed approach. The newly identified craters were attained with low FPR, i.e., $4.49 \pm 0.7\%$ and $4.67 \pm 2.1\%$ for small and large sizes, respectively, which demonstrates that our approach can be used to produce high-precision lunar crater catalogues. Regarding the ages of craters, 88.97 and 89.04% of the estimated craters were assigned to their corresponding formation systems with CE-1 and CE-2 data, respectively, by learning the

morphological features and stratigraphic information. From comparisons with existing lunar age chronologies, most craters have been shown to have good consistency in terms of both relative age and absolute age.

As mentioned in the multiscale crater identification section, missed craters can be avoided to a certain extent by using two types of Chang'E data with different scales and overlapping images. From the detection results, large craters ($D \geq 50$ km) exhibited good agreement with existing crater databases and catalogues. Furthermore, some faint and heavily degraded craters, which are often hard to detect with automated methods, appear in our database. However, the identification is obviously incomplete, as many medium and small craters ($D < 50$ km) are missing from our database compared with the Robbins[11] database. This depends on three main reasons. First, the size of small craters in the feature map of the last convolutional layer is too small for accurate detection. Thus, the two spatial resolutions of the data, i.e., CE-1 120 and CE-2 50 m, are not enough to deal with the vast presence of craters. In further research, we plan to transfer the detection approach to the CE-2 global lunar terrain product with 20 and 7 m spatial resolutions for identifying smaller craters and to build a more complete lunar crater database. Moreover, our model was able to understand the training objective and correlate the binary ring targets with the true rims of the craters with a limited amount of training data. However, the distribution of recognized craters used is incomplete, thus affecting the complete learning of crater features.

The age of craters, i.e., the formation system estimated in our database, is comprehensively estimated by considering morphological features and stratigraphic information. The CNN approach works on windows of pixels. It does not recognize the actual size of craters from the image. Thus, the morphological features were extracted from DOM images of Chang'E data and the calculated morphological data of craters. The stratigraphic features were mapped to corresponding craters. Then, integrated features were learned to contribute to the estimation of craters by a dual-channel crater classification model. The need for integrated feature learning was tested by an ablation study[48], which refers to removing some features of the model and analyzing performance. We applied the classification model with only stratigraphic features by removing the morphological data and obtained an OA of 72.21 ± 3.79% and the best OA of 75.74% on the CE-1 test set of the dated craters. Then, this model was transferred to CE-2, resulting in a 77.09% OA. Compared with the proposed classification model with both morphology and stratigraphic information, the OAs of integrated features are 13.24 ± 2.67% and 11.95% higher than those obtained with only stratigraphic features. These results highlight the robustness and flexibility of crater classification solutions. However, it is worth noting that due to the difference and incompleteness of dated craters from the LPI with stratigraphic information, some limitations exist in the crater age estimation. For example, the crater density in the Eratosthenian System may be underestimated because of the availability of dated samples. More precise stratigraphic data should be determined in future research to guarantee the universality of this method.

In conclusion, a new lunar crater database of 117,240 craters of $D \geq 1$ km (the highest quantity with a low FPR among existing automated catalogues) and 18,996 craters of $D \geq 8$ km with ages (13 times higher than that in existing dated crater datasets) throughout the mid- and low-latitude regions of the Moon has been derived and made available. Additionally, the adopted progressive TL strategy can be applied to assist human crater studies, in particular generating reliable suggestions when facing limited samples in planetary research. This progressive TL strategy implemented in deep architecture is similar to a supervisor passing his/her knowledge (feature representation) and experience (detection and classification capability) from one generation to another. The complex features can be learned identically for every crater to ensure consistency of decisions. The student-like learned model (e.g., the two-stage crater detection approach) can be adapted to other Solar System bodies, e.g., Mars, Mercury, Venus, Vesta and Ceres, to extract much more semantic information than the usual manually analysis method. This prediction will generally take minutes followed by a few hours of post-processing on standard computation hardware.

## Methods

**Datasets**. The data used in this paper come from the CE-1 and CE-2 orbiters of the CLEP, the IAU, the LPI and the USGS. The DOM and DEM data from the CCD stereo cameras aboard CE-1 and CE-2. The 120 and 50 m/pixel images and DEM data were produced by the CE-1[32] and CE-2[33] CCD stereo cameras at orbital altitudes of 200 and 100 km, respectively. The recognized lunar craters come from the IAU[16]. The crater dataset was downloaded on June 17, 2018, and contains 9137 craters. The craters with constrained ages are derived from the LPI document published in 2015 (ref. [17]), which includes 1675 craters with constrained ages. The craters span five lunar geologic periods, i.e., the pre-Nectarian System, the Nectarian System, the Imbrian System (the Lower Imbrian Series and the Upper Imbrian Series), the Eratosthenian System and the Copernican System. The lunar geological map is derived from 1:5,000,000 Lunar Geologic Renovation (2013 edition) produced by the U.S. Geological Survey (https://astrogeology.usgs.gov/search/map/Moon/Geology/Lunar_Geologic_GIS_Renovation_March2013).

**Problem extension**. Impact craters, as the most dominant lunar surface features, can usually be identified according to their near-circular depression structure, and their ages are further estimated from morphological markers and stratigraphic control[19]. To identify craters and estimate their ages automatically, we convert the identification of craters into a target detection task and the age estimation of craters into a taxonomy structure.

**Data consolidation**. The study area ranges from −65° to 65° in latitude and from −180° to 65°, and 65° to 180° in longitude on the Moon. The DOM data and DEM data are projected by Mercator at 33°. For the crater detection task, the DEM data of CE-1 and CE-2 were resampled to 120 and 50 m/pixel. The slope information and the profile curvature were extracted from the DEM data. DOM data and DEM data were fused. To train the crater detection model, the craters in the study area were marked based on the lunar dataset published by IAU[16]. At the same time, the CE-1 data were cut into 5000 × 5000 pixel and 1000 × 1000 pixel images (two different projection coordinates), and CE-2 data were cut into 1000 × 1000 pixel images to detect multiscale craters in the whole study area. To avoid incomplete craters caused by clipping, the adjacent images have a 50% overlap rate.

For classification of craters, we extracted the recognized and identified craters with CE-1 and CE-2 DOM data. Forty morphological information of the craters (e.g., diameter (km), rim to floor depth (km) and interior volume (km³)) were calculated with Chang'E data with reference to the lunar impact crater database published by the LPI[17], and 38 stratigraphic attributes of craters were extracted from the 1:5,000,000 Lunar Geologic Renovation (2013 edition) produced by the U.S. Geological Survey. Finally, 78 attributes of craters were constructed by integrating the morphological features with the corresponding stratigraphic attributes. In the experiments, the craters images with all these attributes are used as the input in deep neural network which can automatically discover the representations needed for classification of craters[37]. Note that the 40 morphological information are the generic morphological parameters of craters, and the 38 stratigraphic attributes are related to the crater materials of the five systems, which are listed in the table of attribute data. Information on these attributes can be found at https://github.com/hszhaohs/DeepCraters/tree/master/age_estimation, together with the documentation of each attribute.

**Sample selection**. For crater detection, combined with the catalogue data published by IAU[16], the CE-1 and CE-2 data yield 7895 and 6511 recognized craters in the study area, respectively. In the detection model, the recognized craters of CE-1 are divided into training and tests set at a ratio of 9:1. During the training process, 20% of the training set is used for validation. All the recognized craters in the CE-2 data are used as the test set.

For estimating crater ages, five classes corresponding to five lunar geologic periods, i.e., the pre-Nectarian System, the Nectarian System, the Imbrian System, the Eratosthenian System and the Copernican System, are considered. A total of 1491 craters with constrained ages in the study area were selected based on the lunar crater database published by the LPI[17]. On the basis of spatial resolution, the CE-1 and CE-2 images contain 1411 and 502 craters with different ages, respectively. To estimate the performance of the classification model and monitor the training process, these craters in the CE-1 data were divided into training, validation and test sets at a proportion of 8:1:1 according to the principle of stratified sampling. All the craters in the CE-2 data are used for testing.

**Crater identification algorithm**. For multiscale crater identification, a two-stage crater detection approach based on TL is designed. The region-based fully convolutional network (R-FCN)[49] is selected as the detection model. First, Resnet101 (ref. [50]) is used as the basic network of the R-FCN detection model to extract features from CE-1 data. The model is initialized with pre-trained parameters on ImageNet[51]. Then, all the parameters of the network are fine-tuned by back propagation using the training craters of CE-1. This is the first-stage TL. The initial learning rate is 0.001. The total number of iterations is set to 100,000, and the learning rate is reduced to 0.0001 when the training iterations reach 80,000. We use the mini-batch stochastic gradient descent optimization method to learn the parameters, in which the batch size is set to 128, the momentum is 0.9, and the weight decay coefficient is 0.0005. The Caffe[52] DL framework is used to train, validate and test the whole detection network. Then, the first-stage crater detection model is obtained for identifying large-scale craters. For the 5000 × 5000 pixel and 1000 × 1000 pixel images, we use the same parameter setting. To identify small-scale craters, we carried out the second stage of the detection approach, i.e., the first-stage crater detection model, which is fine-tuned on CE-1 data, is directly transferred to CE-2 data without any training. The two-stage crater detection approach is implemented with a PC workstation (Intel(R) Core(TM) i7-5930K CPU at 3.50 GHz with 128 GB of RAM and NVIDIA GeForce GTX TITAN X Graphics Processing Unit).

**Identified accuracy metrics**. To evaluate our crater identification algorithm, recall (R) is used to measure the performance of detection and can be calculated as follows:

$$R = \frac{T_p}{T_p + F_n},$$

where $T_p$ is true positives and $F_n$ is false negatives.

In the calculation process, we define detected craters as those that have intersection over union (IoU) overlap with a ground-truth box of at least 0.5; otherwise, they are considered undetected.

**Age estimation algorithm**. Considering the small number of available craters with a constrained age, a two-stage crater classification approach based on TL is proposed. For two types of lunar data, i.e., DOM and attribute data, a dual-channel crater classification model is constructed. One of the channels utilizes a classical deep CNN to extract information from the image features of craters. The other uses a feedforward neural network to assess the morphological and stratigraphic features of craters of different ages. Finally, the two types of features obtained from the two channels are merged for classification. A large number of newly identified craters are used as unlabelled data. A semi-supervised learning strategy (i.e., Meanteacher[53]) is adopted to prevent overfitting[37] of the classification model. To ensure the quality of the unlabelled data, we select identified craters with a confidence level of 0.99 as the input data. In the first stage of the classification approach, the network is initialized by pre-trained parameters on ImageNet[51]. Then, the CE-1 dated craters and the identified craters are fed into the classification model to fine-tune all the parameters. In this model, the image size is set to 256 × 256 pixels (i.e., the craters DOM are resampled for network size). In the training process of the first-stage model, all parameters are fine-tuned by back propagation and trained by the Adam[54] optimizer; the learning rate is 0.0003, the total number of epochs is set to 10, the batch size is set to 32 and the weight attenuation coefficient is 0.0001. The PyTorch DL framework (https://pytorch.org/) is used to train, validate and test the classification network. In the second stage of the classification approach, the model trained by CE-1 data is directly transferred to CE-2 without training and learning. To improve the classification performance of craters, 12 deep CNN models were utilized to extract crater features, i.e., Resnet50 (ref. [50]), Resnet101(ref. [50]), Resnet152 (ref. [50]), Senet[55], se_Resnet50 (ref. [55]), se_Resnet101 (ref. [55]), se_Resnet152 (ref. [55]), se_Resnext101 (ref. [55]), Polynet[56], Inceptionv3 (ref. [57]), DPN68b[58] and Densenet201 (ref. [59]). Thus, we obtain 12 age classification results for the craters. For the CE-1 and CE-2 data, the ensemble strategy is based on a genetic algorithm for weighting the 12 age classification results. The age that receives the highest weighted sum is selected as the final age of the crater.

**Evaluation of crater age classification**. In the age estimation algorithm, the OA is used to analyse the crater classification results, and the confusion matrix is given to show the classification situation of craters with different ages in an overall way. The OA is defined as:

$$OA = \frac{1}{N} \sum_{i=1}^{C} x_{ii},$$

where $N$ is the total number of craters, $C$ represents the number of categories, and $x_{ii}$ is the number of correctly classified craters of the $i$th class.

**Quantification and statistical analysis**. The lunar impact crater age estimation experiments were repeated five times independently. In each trial, 10% craters with constrained ages were independently selected by stratified sampling as the test set. The OA represents the mean ± s.d.

## Data availability

The lunar craters datasets of recognized craters and craters with ages are available from the IAU (https://planetarynames.wr.usgs.gov/Page/MOON/target) and the LPI (https://www.lpi.usra.edu/lunar/surface/), respectively. The Chang'E data used in the experiment are available from the Data Publishing and Information Service System of the CLEP (http://moon.bao.ac.cn). The databases of identified and dated craters are available in figshare with the identifier [https://doi.org/10.6084/m9.figshare.12768539.v1]. The experimental data (including the catalogues of craters and attribute data used to train, validate and test the TL-based strategy) are available at https://github.com/hszhaohs/DeepCraters. Source data are provided with this paper.

## Code availability

The models of crater identification and age estimation are publicly available at https://github.com/hszhaohs/DeepCraters.

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

## Acknowledgements

This study was supported by the CE-1 and CE-2 missions of the CLEP. We thank the team members of the Ground Application and Research System (GRAS), who contributed to data receiving and preprocessing. This work was also supported by the National Natural Science Foundation of China grants 61572228 and 61972174, the Open Funds Project of Key Laboratory of Lunar and Deep Space Exploration LDSE201906, the Science-Technology Development Plan Project of Jilin Province of China grants 20190303006SF and 20190302107GX and the Industrial Innovation Special Funds Project of Jilin Province grants 2019C053-5 and 2019C053-7. The authors are grateful to Dr. Zhiyong Xiao from Sun Yat-sen University for providing helpful comments on a draft of the manuscript and useful analysis of crater populations. The authors thank Stuart J. Robbins for providing the *D* >1 km database, R. Povilaitis and the LROC team for providing the 5–20 km database, James W. Head for providing the *D* >20 km database and Goran Salamunićcar, Weiming Cheng, Ari Silburt for providing the automated crater catalogues. The authors would like to acknowledge Dr. Liang Chen from Shantou University for improving the figures in this paper.

## Author contributions

C.Y., H.Z. and R.G. designed the research and implemented transfer learning algorithms and wrote the manuscript. C.L. and Z.O. contributed scientific background, geological interpretation, and consistency of remote-sensing observations. B.L. and X. Z. conducted data preprocessing. L.B., J.B. and Y.L. provided the background knowledge of models, helped to focus the relevance of the contribution and aided in revising the manuscript.

## Competing interests

The authors declare no competing interests.
