## [Peer Review File · Nature Communications]

Reviewers' comments:

Reviewer #1 (Remarks to the Author):

This paper provides an interesting extraction and aging procedure for the lunar craters using the digital orthophotos and the Digital Elevation Models obtained by the two Chinese lunar missions Chang'E 1 and 2. The paper is interesting from two points of view. On the one side, for the novel approach and idea to apply Deep Learning (DL) techniques to the extraction of lunar craters at multiple scales. On the other side, because the extraction includes a full range of new craters that can be used to extrapolate the history of the Moon and of the solar system.

Because of my background in remote sensing data analysis, I will be able to comment only on the first point and, accordingly, I feel that my review does not provide a full coverage of the paper, because I cannot (and won't) comment on the analysis of the extracted data from the geological point of view. This second part of the paper is apparently the one that the authors consider as more important, because it is described in the main text of the paper, while the crater extraction and aging procedures are described in the additional documents.

wherefore, and specifically discussing the use of DL techniques for crater detection, the authors state that they have used a standard DL architecture, that has been trained using Chang'E-1 data and then applied also to Chang'E-2 data, which have a different spatial resolution. The application to these two data sets allows the extraction of craters with very different diameters, from hundreds of km down to 1 km. This is very interesting. What is not in line with the state of the art is that the authors call this approach "transfer learning". In this case, since the two data sets are essentially different only because of the scale, "transfer learning" means just "change of the scale (spatial resolution) parameter", and since the DL approach works on windows of pixels and do not recognize the actual size of the pixel (in a geographical sense), I feel there is no transfer learning at all. In reality, the same DL architecture is applied to two data sets that correspond to differently spatially sampled images of the same area with the more or less the same sensor. I know this is a simplified statement, but the differences between the two sensors that do not fit into this scheme are minor one, and therefore there is no actual transfer learning at all, in my opinion.

A second point of the paper that is not satisfactory is the description of the features that are in input to the DL architecture, and their meaning. The authors state that they use 40 morphological and 28 stratigraphic information, but they do not offer any table explaining which are these features, do not explain why they were selected, or how these features are extracted from the remotely sensed data, and whether they also use geological maps of the Moon as input and complementary information to Chang'E data. Instead, everything very vague, with only a few examples of features mentioned for morphological ones, and no one for geological ones. Stating that there are so many features but without offering any information about them makes this paper absolutely impossible to replicate, hence not useful as scientific literature.

Additionally, the results show large recall percentages, which means that the DL approach is able to recognize almost all of the craters that are used as test set. Despite these numbers are very good, this is not a proof of the excellence of the technique, because the proposed technique eventually extracts more than 10 times as many craters as the previously known ones, hence the reported positive numbers (more than 90% recall) might be just due to an over extraction of craters all over the Moon surface. Honestly, I do not think this is the case, but I would like to use this example to stress that reporting recall percentages alone, in this case, it is not a statistically significant performance assessment technique.

As a side note, I should add that I do not like the idea that the authors of this paper will provide the results only to interested parties upon request. I would advocate for a free distribution of the extracted and aged crater database. Since it is so vast, it makes sense to let everybody work on it so that the crater extraction can be used, validated, improved and shared, and the lunar exploration will profit much more than by the work of just a few researchers who are going to ask for them directly to the authors, and eventually receive them.

Reviewer #2 (Remarks to the Author):

Review by Clark R. Chapman (not anonymous) of a paper, submitted to Nature Communications, by Chen Yang et al. "Lunar impact craters identification and age estimation with Chang'E data by deep and transfer

learning.”

The authors are to be congratulated in attempting to apply modern machine learning methods to identifying craters from the comprehensive imaging data of the two Chang'E missions to the Moon. Certainly I hope that the results of this project can eventually be published. However, there are some serious issues that must be addressed before publication. It is possible that the authors have omitted reporting some critical details of their lunar-science methodology, or perhaps they need to look more deeply into the lunar crater literature. A colleague of mine who could help them is Dr. Zhiyong Xiao at Sun Yat-sen University (xiaozhiyong@mail.sysu.edu.cn). Let me explain.

This paper attempts an interdisciplinary task: to contribute to lunar and solar system science by analyzing the innumerable craters on the Moon using modern deep learning methodologies that have remarkably advanced in recent years. Although I was an early researcher to apply pattern recognition methods to identification of lunar and planetary craters, the advances since then have been enormous and I am not an expert in the current state-of-the-art. Thus I cannot critique those methodologies except noting (e.g. from Extended Data Fig. 2) that craters appear to be reliably identified.

Unfortunately, no matter how excellent the machine-learning software, if the training dataset is poor or not understood, the results of the automated crater identifications – and especially the assigned “ages” based on morphological attributes of the craters – cannot be reliable. The authors appear to be unaware that instead of there being less than 10,000 recognized craters on the Moon, there are many other lunar crater databases that provide positions and diameters for up to 2 million craters. Other databases with intermediate numbers of craters also provide morphological classifications. Instead, these authors appear to use for the training set only a portion of a bibliographical database of just *named* craters. Craters have been named, or not, for reasons having little to do with science. Moreover, the USGS database that the authors presumably used (though their citation is incomplete, so I can't be sure) has 5 columns (mostly very incomplete) for crater “ages” or assignments to lunar systems. The authors do not discuss the criteria they used for selecting an age, and the database itself does not critically analyze the methods or reliability of the age estimates.

Most of the vast literature on lunar crater chronology uses methodologies not used in the present paper, tying crater densities to ages determined from returned lunar samples. Even age estimates of individual multi-km diameter craters, as attempted in this paper, use statistics of overlapping small craters to determine ages (e.g. \bibitem[Kirchoff et al.(2013)] Kirchoff, M.~R., Chapman, C.~R., Marchi, S., Curtis, K.~M., Enke, B., Bottke, W.~F. \ 2013. \ Ages of large lunar impact craters and implications for bombardment during the Moon's middle age. \ Icarus 225, 325). In the present paper, it is attempted only to assign a crater to one of five system epochs on the Moon, and it is apparently based on the morphology of the crater. While it is true that there is some correlation between crater morphology and age, there is a very strong dependence on crater size and other factors. And given the very non-representative nature of “named” lunar craters (which are named because of attributes highly correlated with morphology and size), the use of such named craters as a training set is very problematical. It is possible that these issues were considered by the authors but I can't find such arguments or discussion in the text.

I am particularly concerned about Fig. 2. Such a plot of cumulative numbers versus size depends on there being no biases due to crater size. Yet obviously there is such a bias in named craters, which I assume are the “previously mapped” or “known” craters referred to in the figure and its caption. And I see no discussion of what I expect are similar biases against smaller craters in the identification algorithms. As a result, the discussion in the text of results based on Fig., 2 has poor reliability, I expect. Of course, the five age categories are so broad that there is presumably some correct qualitative result, but not results that are useful for understanding “the dynamical history of the Solar System,” as the authors claim. Let me add that there is considerable debate about the ages of the boundaries of the five systems and also whether or not there was a Late Heavy Bombardment, which the authors not only seem to believe is proved, but which they claim to have further proved.

I hope that the authors can learn a little bit more from the extensive literature on lunar crater statistics and

reapply their machine learning methodologies in a way that can be more useful for understanding solar system cratering chronologies. This paper is not yet ready for publication before considerable revision and augmentation. I try to append a copy of the text which I have marked up and commented on using Track Changes.

Reviewer #3 (Remarks to the Author):

The manuscript uses transfer learning to train deep neural networks to identify and date an order of magnitude more small craters than have been identified to date using data from the Chang'E orbiters. This would represent an important new dataset potentially encoding new insights into the early history of the Moon and inner solar system. It also employs techniques applicable across disciplines that would be of wide interest to readers of nature communications, so I believe it potentially merits publication in the journal.

However, I have reservations about the statistical validation of the model. Partly because of such a dataset's value, I believe there is a strong burden of proof, which I do not feel the manuscript currently meets. I outline below some possible tests, or perhaps the authors can develop even better ones.

I also do not feel like the scientific conclusions in the manuscript are warranted, as expanded on below. I don't think this is fatal—a public dataset with an order of magnitude more craters would stand on its own for its scientific significance to the field. I would recommend the scientific inferences be removed or at least softened before publication.

Scientific Conclusions:

The trained model finds a large number of small craters, and assigns them to five different age systems. The manuscript qualitatively describes the various features in the inferred size distribution in Fig 2b (which is appropriate), but then connects some of them directly to events in the solar system, e.g. "It is consistent with and further proves the LHB occurred on the Moon." Connecting these size distributions to particular events in the solar system's history is a complicated undertaking with a large literature. Especially for the small new craters the model identifies, the strong erosion, which would differ between craters of different age systems makes this connection particularly challenging. The manuscript does not adequately justify the connection, and may not even be realistic in the space of this letter. Discussion of the qualitative trends, perhaps with more tentative inferences and discussion of caveats, seems more appropriate. As stated above, my feeling is that the dataset and methods can stand on their own.

Statistical Analysis:

The value of this crater dataset imposes a strong burden of proof. The main product of this work is a dataset of small new craters, but the current manuscript does not adequately assess the reliability of these craters, particularly at small sizes. I think incorporating more testing should be a minimum requirement for publication.

The paper quotes the models' recall, but shouldn't this trade off against its precision? Presumably the model's precision is an important metric for people trying to extract scientific conclusions from the dataset, I.e., given this dataset of craters, how many of them are actually real? Of course this is complicated by the fact that most of the found craters are newly identified. However, it would seem that one could impose a minimum crater size, above which we can be confident the crater counts are complete, and ask what the precision is. Some analysis of what objects confused the model and were incorrectly labeled as craters would significantly strengthen the results.

It would also be valuable to find a way to actually verify the new small craters that the model is finding.

Perhaps one could calculate the precision in bins of different crater sizes, down to a size where the crater counts should be complete? If the precision stayed constant (or improved) toward lower crater sizes, that would give some confidence in the dataset beyond the point where crater counts stop being complete. Alternatively, perhaps one could have an expert identify objects below the training dataset size threshold, both ones that are craters with high confidence and non-craters with high confidence.

It would be useful to overplot the known craters in different age systems in Fig 2b, as is done in the left panel. It seems like the breaks in power law slopes in Fig 2b could be close to the minimum size of known craters. If the observed power law breaks really are close to the minimum sizes in the training data, that would seem to undermine the validity of the results. Or perhaps there is an important physical distinction at that particular size scale, which is why crater counters stop at that size, but there needs to be some discussion around this if it is true that the power law breaks are close to the minimum sizes in the training set.

I was confused by the five trials on the age classification problem. Are the quoted accuracies not done on the single holdout set? Are these the performance on five-fold cross validation? If so, what about the performance on the holdout set never seen by the model? Again the statistical analysis here does not consider the effect of the adopted thresholds for classification. Shouldn't there be a tradeoff between precision and recall (see above)? It would be helpful to the reader to analyze the crater identification and age system classification in similar ways. Is there a reason why accuracy is quoted here, while recall is quoted earlier?

Is there a phase angle dependence for the DOM data? Are there images of the same fields at different times/phase angles? If so, how much do the probabilities of being a crater and belonging to various system ages vary for the same crater in different images?

Minor comments:

The age system classifier uses a CNN and engineered features in parallel, weighting outputs from 12 different CNNs...this seems like quite a complicated model. Can you give a sense for how much the CNNs help? Presumably the engineered features were added to the pure CNN architectures used earlier in order to help the performance—but how much does the performance degrade if ONLY the engineered features are used?

At several points, the manuscript says 1675 determined ages are obviously not satisfactory. It should identify a specific task and justify why they're not satisfactory for that task, or rephrase

Reviewer(s) Comments:

Reviewer: 1

This paper provides an interesting extraction and aging procedure for the lunar craters using the digital orthophotos and the Digital Elevation Models obtained by the two Chinese lunar missions Chang' E 1 and 2. The paper is interesting from two points of view. On the one side, for the novel approach and idea to apply Deep Learning (DL) techniques to the extraction of lunar craters at multiple scales. On the other side, because the extraction includes a full range of new craters that can be used to extrapolate the history of the Moon and of the solar system.

Because of my background in remote sensing data analysis, I will be able to comment only on the first point and, accordingly, I feel that my review does not provide a full coverage of the paper, because I cannot (and won't) comment on the analysis of the extracted data from the geological point of view. This second part of the paper is apparently the one that the authors consider as more important, because it is described in the main text of the paper, while the crater extraction and aging procedures are described in the additional documents.

1. Therefore, and specifically discussing the use of DL techniques for crater detection, the authors state that they have used a standard DL architecture, that has been trained using Chang'E-1 data and then applied also to Chang'E-2 data, which have a different spatial resolution. The application to these two data sets allows the extraction of craters with very different diameters, from hundreds of km down to 1 km. This is very interesting. What is not in line with the state of the art is that the authors call this approach "transfer learning". In this case, since the two data sets are essentially different only because of the scale, "transfer learning" means just "change of the scale (spatial resolution) parameter", and since the DL approach works on windows of pixels and do not recognize the actual size of the pixel (in a geographical sense), I feel there is no transfer learning at all. In reality, the same DL architecture is applied to two data sets that correspond to differently spatially sampled images of the same area with the more or less the same sensor. I know this is a simplified statement, but the differences between the two sensors that do not fit into this scheme are minor one, and therefore there is not actual transfer learning at all, in my opinion.

Reply: Thank you for your comments and suggestion. There are three reasons for considering this application as transfer learning when using Chang'E-1 (CE-1) data and Chang'E-2 (CE-2) data to identify and estimate the age of different scale craters. First, although the CE-1 and CE-2 orbiters belong to the China's Lunar Exploration Program (CLEP), some parameters, e.g. the satellite track heights, sensor's viewing angles and spatial resolution determine significant differences between the CE-1 and CE-2 data. The satellite track heights of the CE-1 and CE-2 are 200 km and 100km, respectively. For the sensor's viewing angles, CE-1 has the normal view, the frontward and the backward views, while CE-2 only has the frontward view and the backward. Two kinds of global lunar terrain products named CE1TMap2009 and CE2TMap2015 with different resolution are produced. Second, for detecting multi-scale craters in the whole study area, CE-1 data are cut into 5000×5000 pixels and 1000×1000 pixels images (two different projection coordinates), and CE-2 data to 1000×1000 pixels images. Given the differences between the CE-1 and CE-2 data, the same crater shows obvious variability in location and size even at the same scale images of CE-1 and CE-2 (see Fig.2a in the revised paper). Third, the available training data are very scarce compared with the large number and types of craters. In CE-1 data and CE-2 data, 7895 and 6511 recognized craters and 1411 and 502 aged craters in the study area can be used in the experiments, respectively. In the first stage approach, the partial networks of existing CNNs trained with

ImageNet data are transferred and reused in CE-1 data, according to a network-based deep transfer learning [1]. In the second stage, we suppose the training data is not available in CE-2, and the first stage detection model is transferred to CE-2 data without any training data. The learning procedure of the second stage follows transductive transfer learning that can provide learned features and knowledge for CE-2 data [2]. This has been claimed in the revised paper (pages 2 and 6).

[1] Tan C., Sun F., Kong T., Zhang W., Yang C. & Liu C. A survey on deep transfer learning. *arXiv* (2018).

[2] Pan, S. J. & Yang, Q. A survey on transfer learning. *IEEE Trans. Knowl. Data Eng.* 22, 1345-1359 (2010).

2. A second point of the paper that is not satisfactory is the description of the features that are in input to the DL architecture, and their meaning. The authors state that they use 40 morphological and 38 stratigraphic information, but they do not offer any table explaining which are these features, do not explain why they were selected, or how these features are extracted from the remotely sensed data, and whether they also use geological maps of the Moon as input and complementary information to Chang' E data. Instead, everything very vague, with only a few examples of features mentioned for morphological ones, and no one for geological ones. Stating that there are so many features but without offering any information about them makes this paper absolutely impossible to replicate, hence not useful as scientific literature.

Reply: Thank you for your comment. The experimental data (including the catalogues of craters and attribute data used to train, validate and test) and the models of craters identification and age estimation are available at <https://github.com/hszhaohs/DeepCraters>. This has been stated in Data availability section of the revised manuscript (page 13).

3. Additionally, the results show large recall percentages, which means that the DL approach is able to recognize almost all of the craters that are used as test set. Despite these numbers are very good, this is not a proof of the excellence of the technique, because the proposed technique eventually extracts more than 10 times as many craters as the previously known ones, hence the reported positive numbers (more than 90% recall) might be just due to an over extraction of craters all over the Moon surface. Honestly, I do not think this is the case, but I would like to use this example to stress that reporting recall percentages alone, in this case, it is not a statistically significant performance assessment technique.

Reply: Thank you for your comment. In the craters identification experiment, the comparison with existing lunar craters databases and false positive rates of the identified craters have been added to the Results section where lunar crater size distribution is compared with existing lunar craters databases (pages 2, 4, 5 and 6). Meanwhile, a description and analysis of the comparison results is given.

4. As a side note, I should add that I do not like the idea that the authors of this paper will provide the results only to interested parties upon request. I would advocate for a free distribution of the extracted and aged crater database. Since it is so vast, it makes sense to let everybody work on it so that the crater extraction can be used, validated, improved and shared, and the lunar exploration will profit much more than by the work of just a few researchers who are going to ask for them

directly to the authors, and eventually receive them.

Reply: Thank you for your suggestion. The Chang' E data used in the experiment is provided in Source Data and available from the Data Publishing and Information Service System of the China's Lunar Exploration Program (<http://moon.bao.ac.cn>). The databases of identified and aged craters are provided in Supplementary Dataset and available at https://figshare.com/articles/dataset/CE_DeepCraters/12768539. The experimental data (including the catalogues of craters and attribute data used to train, validate and test) and models of craters identification and age estimation is available at <https://github.com/hszhaohs/DeepCraters>. It has been stated in the Data availability section (page 13).

Reviewer: 2

Review by Clark R. Chapman (not anonymous) of a paper, submitted to Nature Communications, by Chen Yang et al. “Lunar impact craters identification and age estimation with ChangE data by deep and transfer learning.”

The authors are to be congratulated in attempting to apply modern machine learning methods to identifying craters from the comprehensive imaging data of the two Chang’ E missions to the Moon. Certainly I hope that the results of this project can eventually be published. However, there are some serious issues that must be addressed before publication. It is possible that the authors have omitted reporting some critical details of their lunar-science methodology, or perhaps they need to look more deeply into the lunar crater literature. A colleague of mine who could help them is Dr. Zhiyong Xiao at Sun Yat-sen University (xiaozyhong@mail.sysu.edu.cn). Let me explain.

1. This paper attempts an interdisciplinary task: to contribute to lunar and solar system science by analyzing the innumerable craters on the Moon using modern deep learning methodologies that have remarkably advanced in recent years. Although I was an early researcher to apply pattern recognition methods to identification of lunar and planetary craters, the advances since then have been enormous and I am not an expert in the current state-of-the-art. Thus I cannot critique those methodologies except noting (e.g. from Extended Data Fig. 2) that craters appear to be reliably identified.

Reply: Thank you very much for your encouragement and suggestions.

2. Unfortunately, no matter how excellent the machine-learning software, if the training dataset is poor or not understood, the results of the automated crater identifications – and especially the assigned “ages” based on morphological attributes of the craters – cannot be reliable. The authors appear to be unaware that instead of there being less than 10,000 recognized craters on the Moon, there are many other lunar crater databases that provide positions and diameters for up to 2 million craters. Other databases with intermediate numbers of craters also provide morphological classifications. Instead, these authors appear to use for the training set only a portion of a bibliographical database of just *named* craters. Craters have been named, or not, for reasons having little to do with science. Moreover, the USGS database that the authors presumably used (though their citation is incomplete, so I can’t be sure) has 5 columns (mostly very incomplete) for crater “ages” or assignments to lunar systems. The authors do not discuss the criteria they used for selecting an age, and the database itself does not critically analyze the methods or reliability of the age estimates.

Reply: Thank you for your comments. In this paper, the recognized lunar craters used for craters identifications is selected from the International Astronomical Union (IAU) (<https://planetarynames.wr.usgs.gov/Page/MOON/target>) and the craters assigned ages from the lunar impact craters database of the Lunar and Planetary Institute (LPI) (<https://www.lpi.usra.edu/lunar/surface/>). The reasons for which we choose these two datasets are as follows: First, it should be acknowledged that many existing lunar crater databases have made great contribution to the scientific research of the Moon. However, the manual subjectivity and the different specific automatic detection priorities with different types of data result in significant disagreement in crater counting, location and size in different databases. The comparison between craters detected in this paper and currently public lunar crater databases was carried out and the analysis of results has been added

in the Results section of the revised paper for lunar crater size distribution (pages 2, 4, 5 and 6). On the other hand, the ages information of craters is scarce. We considered the formation Systems of craters aggregated by the LPI, according to Wilhelms' professional paper from the United States Geological Survey (USGS) based on the geologic history of the Moon and updated with the stratigraphy of lunar craters database (which is at present a comprehensive database for craters ages). The reason and necessity of utilization of the training data has been described and added in the introduction section (page 2).

Therefore, the purpose of this paper is to use a small number of training data to identify craters effectively and estimate crater age accurately from the same source of data and with the same objective automatic approach. Experimental results show that the proposed transfer learning strategy with a small number of training craters can extract much more craters than when using only a deep learning-based method with a large number of training samples. In the craters age estimation algorithm, two types of input data have been used, i.e., images and 78 attribute data, including 40 morphological features of craters (e.g. diameter [km], rim to floor depth [km], interior volume [km³]) were calculated with Chang'E data referring to the lunar impact craters database published by LPI and 38 stratigraphic information of craters were extracted from the 1:5,000,000 Lunar Geologic Renovation (2013 edition) produced by the U.S. Geological Survey. The experimental data (including the catalogues of craters and attribute data used to train, validate and test) and the models of craters identification and age estimation are available at <https://github.com/hszhaohs/DeepCraters>. This is stated in the Data availability section of the revised paper (page 13).

3. Most of the vast literature on lunar crater chronology uses methodologies not used in the present paper, tying crater densities to ages determined from returned lunar samples. Even age estimates of individual multi-km diameter craters, as attempted in this paper, use statistics of overlapping small craters to determine ages (e.g. Kirchoff et al.(2013) Kirchoff, M.-R., Chapman, C.-R., Marchi, S., Curtis, K.-M., Enke, B., Bottke, W.-F. 2013. Ages of large lunar impact craters and implications for bombardment during the Moon's middle age. *Icarus* 225, 325). In the present paper, it is attempted only to assign a crater to one of five system epochs on the Moon, and it is apparently based on the morphology of the crater. While it is true that there is some correlation between crater morphology and age, there is a very strong dependence on crater size and other factors. And given the very non-representative nature of "named" lunar craters (which are named because of attributes highly correlated with morphology and size), the use of such named craters as a training set is very problematical. It is possible that these issues were considered by the authors but I can't find such arguments or discussion in the text.

Reply: Thank you for your comments. As already mentioned, for lunar impact craters ages estimation, two types of input data, i.e., images and 78 attribute data, including 40 morphological features of craters (e.g. diameter [km], rim to floor depth [km], interior volume [km³]) were extracted from Chang'E data referring to the lunar impact craters database published by LPI and 38 stratigraphic information of craters were extracted from the 1:5,000,000 Lunar Geologic Renovation (2013 edition) produced by the U.S. Geological Survey. The description and analysis of the results of craters classification have been reorganized and rewritten in the Results section for aged lunar impact craters distribution and mapping (pages 7, 8 and 9). Meanwhile, an analysis of consistency with existing lunar age chronology has been added, including the relative age using

the optical maturity and the absolute age obtained by analyzing the craters size-frequency distributions (CSFDs) and the thermophysical characteristics of lunar impact ejecta (pages 9, 10 and 11).

The necessity of integrated features learning is tested using an ablation study, which refers to removing some “features” of the model and analyzing how that affects performance. We designed the classification with only stratigraphic features by removing the morphological features and obtain an overall accuracy (OA) of $72.21\% \pm 3.79\%$ and the best OA of 75.74% on the CE-1 test set of the aged craters; the OA of CE-2 is 77.09% . Compared with the proposed classification model with both morphology and stratigraphic information, the OAs of integrated features are $13.24\% \pm 2.67\%$ and 11.95% higher than those obtained with only stratigraphic features. These results highlight the robustness and flexibility of craters classification solutions.

4. I am particularly concerned about Fig. 2. Such a plot of cumulative numbers versus size depends on there being no biases due to crater size. Yet obviously there is such a bias in named craters, which I assume are the “previously mapped” or “known” craters referred to in the figure and its caption. And I see no discussion of what I expect are similar biases against smaller craters in the identification algorithms. As a result, the discussion in the text of results based on Fig., 2 has poor reliability, I expect. Of course, the five age categories are so broad that there is presumably some correct qualitative result, but not results that are useful for understanding “the dynamical history of the Solar System,” as the authors claim. Let me add that there is considerable debate about the ages of the boundaries of the five systems and also whether or not there was a Late Heavy Bombardment, which the authors not only seem to believe is proved, but which they claim to have further proved.

Reply: Thank you for your input. Fig. 2a has been replaced with the histogram of identified and recognized craters from different diameter scales, see Fig. 3a in the revised paper (page 5). The discussion of distributions of identified craters has been rewritten in the Results section taking into account the precious comments of the Reviewer for lunar crater size distribution and comparison with existing lunar craters databases (page 2). The experiment for lunar craters age estimation has been reorganized. The description and analysis of the results of craters age estimation have been also rewritten in the Results section related to aged lunar impact craters distribution and mapping (pages 7, 8 and 9).

5. I hope that the authors can learn a little bit more from the extensive literature on lunar crater statistics and reapply their machine learning methodologies in a way that can be more useful for understanding solar system cratering chronologies. This paper is not yet ready for publication before considerable revision and augmentation. I try to append a copy of the text which I have marked up and commented on using Track Changes.

Reply: Thank you for your suggestion. The related experiments for lunar craters identification and age estimation have been reorganized, improved and the analysis has been rewritten. In the revised paper, the comparison with existing lunar craters databases and false positive rates of the identified craters have been presented and added in the Results section for lunar crater size distribution and comparison with existing lunar craters databases (pages 2, 4, 5 and 6). The experiments represent up to 85.68% of agreement with published lunar craters databases, and the false positive rate of craters detected is $1.99\% \pm 0.34\%$. For lunar impact craters ages estimation,

the description and analysis of the results of craters classification have been reorganized and rewritten in the Results section related to aged lunar impact craters distribution and mapping (pages 7, 8 and 9). The analysis of consistency with existing lunar age chronology has been added, including the relative age using optical maturity and the absolute age obtained by analyzing the craters size-frequency distributions (CSFDs) and the thermophysical characteristics of lunar impact ejecta (pages 9, 10 and 11). From comparison with existing lunar age chronology, most craters have been illustrated to have good consistency with both relative age and absolute age aspects. However, it is worth noting that due to the difference and incompleteness of aged craters from the LPI with stratigraphic information, some limitations exist in the crater age estimation. For example, the craters density in the Eratosthenian System may be underestimated as greatly influenced by the aged available samples. More precise stratigraphic data should be determined in future research for guarantying the universality of this method. This is stated in the Discussion of the revised paper (page 12).

Reviewer: 3

The manuscript uses transfer learning to train deep neural networks to identify and date an order of magnitude more small craters than have been identified to date using data from the Chang' E orbiters. This would represent an important new dataset potentially encoding new insights into the early history of the Moon and inner solar system. It also employs techniques applicable across disciplines that would be of wide interest to readers of nature communications, so I believe it potentially merits publication in the journal.

1. However, I have reservations about the statistical validation of the model. Partly because of such a dataset's value, I believe there is a strong burden of proof, which I do not feel the manuscript currently meets. I outline below some possible tests, or perhaps the authors can develop even better ones.

Reply: Thank you very much for your encouragement and suggestions. The related experiments for lunar craters identification and age estimation have been reorganized and added, and the analysis and discussion have been rewritten. The major changes are summarized as follows: *i*) For the identification of lunar impact craters, the comparison with existing lunar craters databases and false positive rates of the identified craters have been considered and added to the Results section (pages 2, 4, 5 and 6). *ii*) For the ages estimation of lunar impact craters, the description and analysis of the results of craters classification have been reorganized and rewritten in the Results section (pages 7, 8 and 9). Meanwhile, an analysis of consistency with existing lunar age chronology has been added, including the relative age using optical maturity and the absolute age obtained by analyzing the craters size-frequency distributions (CSFDs) and the thermophysical characteristics of lunar impact ejecta (pages 9, 10 and 11).

2. I also do not feel like the scientific conclusions in the manuscript are warranted, as expanded on below. I don't think this is fatal—a public dataset with an order of magnitude more craters would stand on its own for its scientific significance to the field. I would recommend the scientific inferences be removed or at least softened before publication.

Reply: Thank you for your suggestion. In the revised paper, the scientific inferences have been removed and softened. The description and analysis of the results of craters age estimation have been rewritten in the Results section (pages 7, 8 and 9).

Scientific Conclusions:

3. The trained model finds a large number of small craters, and assigns them to five different age systems. The manuscript qualitatively describes the various features in the inferred size distribution in Fig 2b (which is appropriate), but then connects some of them directly to events in the solar system, e.g. “It is consistent with and further proves the LHB occurred on the Moon.” Connecting these size distributions to particular events in the solar system's history is a complicated undertaking with a large literature. Especially for the small new craters the model identifies, the strong erosion, which would differ between craters of different age systems makes this connection particularly challenging. The manuscript does not adequately justify the connection, and may not even be realistic in the space of this letter. Discussion of the qualitative trends, perhaps with more tentative inferences and discussion of caveats, seems more appropriate. As stated above, my feeling is that the dataset and methods can stand on their own.

Reply: Thank you for your input. In the revised paper, the description and analysis of the

distributions of the aged lunar impact craters have been rewritten in the Results section taking into account your suggestion (page 8).

Statistical Analysis:

4. The value of this crater dataset imposes a strong burden of proof. The main product of this work is a dataset of small new craters, but the current manuscript does not adequately assess the reliability of these craters, particularly at small sizes. I think incorporating more testing should be a minimum requirement for publication.

Reply: Thank you for your comment. In the revised paper, the comparison with existing lunar craters databases and the false positive rates of the identified craters has been considered and added to the Results section (pages 2, 4, 5 and 6). The experimental results show that this work generally achieve high agreement (93.66%) with craters present in other databases for craters $D \geq 3$ km, whereas for smaller diameters there is significant disagreement.

For the ages estimation of lunar impact craters, the analysis of consistency with existing lunar age chronology has been added, including the relative age using optical maturity and the absolute age obtained by analyzing the craters size-frequency distributions (CSFDs) and thermophysical characteristics of lunar impact ejecta for different craters scale (pages 9, 10 and 11). From comparison with existing lunar age chronology, most craters have been illustrated to have good consistency with both relative age and absolute age aspects.

5. The paper quotes the models' recall, but shouldn't this trade off against its precision? Presumably the model's precision is an important metric for people trying to extract scientific conclusions from the dataset, i.e., given this dataset of craters, how many of them are actually real? Of course this is complicated by the fact that most of the found craters are newly identified. However, it would seem that one could impose a minimum crater size, above which we can be confident the crater counts are complete, and ask what the precision is. Some analysis of what objects confused the model and were incorrectly labeled as craters would significantly strengthen the results.

Reply: Thank you for your suggestion. As mentioned above, in the revised paper, the comparison with existing lunar craters databases and the false positive rates of the identified craters have been considered and added to the Results section (pages 2, 4, 5 and 6). The experimental results show that our work generally achieve high agreement (93.66%) on craters present in other databases for crater diameter $D \geq 3$ km. For smaller diameters, there is a significant disagreement. This might be due to the crater detection mechanism in CNN that is based on rectangular shape and it is difficult to guarantee scaling to small crater diameters. Thus, the classification model is utilized to assign ages to identified and recognized craters having diameters larger than 8km (for small craters degrade at an accelerated with respect to large craters of the same age).

6. It would also be valuable to find a way to actually verify the new small craters that the model is finding. Perhaps one could calculate the precision in bins of different crater sizes, down to a size where the crater counts should be complete? If the precision stayed constant (or improved) toward lower crater sizes, that would give some confidence in the dataset beyond the point where crater counts stop being complete. Alternatively, perhaps one could have an expert identify objects below the training dataset size threshold, both ones that are craters with high confidence and

non-craters with high confidence.

Reply: Thank you for your suggestion. The comparison with existing three manual and three automated lunar craters databases in matching number and percentage from different diameters scales have been considered and presented in the Results section (pages 2, 4, 5 and 6). The comparison analysis with public craters databases for different crater sizes has been added. Then, the false positive rates of the identified craters have been given (page 6).

7. It would be useful to overplot the known craters in different age systems in Fig 2b, as is done in the left panel. It seems like the breaks in power law slopes in Fig 2b could be close to the minimum size of known craters. If the observed power law breaks really are close to the minimum sizes in the training data, that would seem to undermine the validity of the results. Or perhaps there is an important physical distinction at that particular size scale, which is why crater counters stop at that size, but there needs to be some discussion around this if it is true that the power law breaks are close the minimum sizes in the training set.

Reply: Thank you for your comment. The CSFDs of the craters with known ages have been supplemented in Fig. 2b, see Fig. 6b in the revised paper (page 8). The analysis of the CSFDs distributions of identified and aged craters has been rewritten in the Results section for aged lunar impact craters distribution and mapping (pages 7, 8 and 9).

8. I was confused by the five trials on the age classification problem. Are the quoted accuracies not done on the single holdout set? Are these the performance on five-fold cross validation? If so, what about the performance on the holdout set never seen by the model? Again the statistical analysis here does not consider the effect of the adopted thresholds for classification. Shouldn't there be a tradeoff between precision and recall (see above)? It would be helpful to the reader to analyze the crater identification and age system classification in similar ways. Is there a reason why accuracy is quoted here, while recall is quoted earlier?

Reply: Thank you for your comment. In the age classification task, five independent trials are carried out to ensure the reliability of age estimation. In each trial, 10% craters with constrained ages are independently selected by stratified sampling as the test set to evaluate the performance of age estimation algorithm. Therefore, the quoted accuracies indicate the mean accuracy of five trials, but not the performance on five-fold cross validation.

For lunar impact craters identification, 791 images including 791 recognized craters are used for test set. However, our craters identification algorithm identifies far more 791 craters from these images, thus the recall is quoted to evaluate the performance of identification algorithm. For the evaluation criteria of age classification, overall accuracy as the common classification evaluation index, is adopted to assess the performance of age estimation algorithm. Moreover, the confusion matrix is also adopted to specify classification performance. This is also the reason for which the accuracy is quoted here. The age estimation in this paper is a multi-classification problem, the crater is classified as the formation System with the highest probability. Regarding the reliability of the identified craters please refers to reply 5 above.

9. Is there a phase angle dependence for the DOM data? Are there images of the same fields at different times/phase angles? If so, how much do the probabilities of being a crater and belonging to various system ages vary for the same crater in different images?

Reply: In the phase of lunar impact craters identification, the same craters can be identified from both CE-1 and CE-2 data. In order to ensure the uniqueness of craters, the repeated craters are removed by reserving $D \geq 20\text{km}$ detected in CE-1 and $D < 20\text{km}$ in CE-2. As a result, the situation of same crater in different images belonging to various system ages will not occur in the phase of lunar impact craters ages estimation.

Minor comments:

10. The age system classifier uses a CNN and engineered features in parallel, weighting outputs from 12 different CNNs...this seems like quite a complicated model. Can you give a sense for how much the CNNs help? Presumably the engineered features were added to the pure CNN architectures used earlier in order to help the performance—but how much does the performance degrade if ONLY the engineered features are used?

Reply: We use a Genetic Algorithm to search the optimal weights of 12 different CNNs. The best performance of different trials is achieved with 7 or 8 CNNs. The need of learning with integrated features is tested using an ablation study which refers to removing some “features” of the model and seeing how that affects performance. We apply the classification model on the CE-1 test set of the aged craters with only stratigraphic features by removing the morphological data and obtained overall accuracy (OA) of $72.21\% \pm 3.79\%$ and the best OA of 75.74% on the CE-1 test set of the aged craters; the OA of CE-2 is 77.09% . Compared with the proposed classification model with both morphology and stratigraphic information, the OAs of integrated features are $13.24\% \pm 2.67\%$ and 11.95% higher than those obtained with only stratigraphic features. These results highlight the robustness and flexibility of craters classification solutions. This is stated in the Discussion of the revised paper (page 12).

11. At several points, the manuscript says 1675 determined ages are obviously not satisfactory. It should identify a specific task and justify why they’re not satisfactory for that task, or rephrase.

Reply: Thank you for your suggestion. The related sentence has been rewritten (page 2).

REVIEWERS' COMMENTS

Reviewer #3 (Remarks to the Author):

I appreciate the authors' efforts to address my concerns and incorporate them into the paper. While I do not feel like their method of estimating the FPR captures the uncertainties in the underlying ground truth, the paper clearly explains their methodology for calculating the FPR.

I am happy to accept to accept the paper, and congratulate the authors for both the impressive technical achievement and for openly sharing it with the scientific community.

Reply to the Reviewers' Comments

The authors would like to thank the referees for the valuable comments and suggestions on this paper. We have carefully and thoroughly revised our paper (Numbered NCOMMS-19-38689-A) in light of the reviewers' comments. A point-by-point response to the comments is provided below. The major changes have been highlighted by colored text in the revised manuscript.

Reviewer(s) Comments:

Reviewer: 1

1. The new version of the paper addresses most of my concerns. I still think that training ImageNet and then fine tuning it with peculiar features is NOT a transfer learning approach, but I concede that moving from CE-1 to CE-2 data may be considered as transfer learning, given the differences (although minor) that the two data sets have.

Reply: Thank you for your comments and suggestion. In the first stage, a partial network of the existing CNNs pre-trained with ImageNet data is transferred and reuses CE-1 data, according to a network-based deep transfer learning [1]. The final classification layer from the network is then retrained with CE-1 data for fine-tuning the parameters across all layers according to ref [2] and [3]. This has been clarified in the revised paper (pages 2 and 6).

[1] Tan C., Sun F., Kong T., Zhang W., Yang C. & Liu C. A survey on deep transfer learning. *International Conference on Artificial Neural Networks, In ICANN*, 270-279 (2018).

[2] Pan, S. J. & Yang, Q. A survey on transfer learning. *IEEE Trans. Knowl. Data Eng.* 22, 1345–1359 (2010).

[3] Esteva, A., Kuprel, B., Novoa, R. A., Ko, J., Swetter, S. M., Blau, H. M. & Thrun, S. Dermatologist-level classification of skin cancer with deep neural networks. *Nature*, 542:115-118 (2017).

2. I am not satisfied however of the reply by the authors about the features used for classifying the craters. The authors state that they are listed in the GitHub repository of the codes, which looked strange to me. Indeed, going to the GitHub repository there is no description, and one need to look inside the code to understand how it work. I strongly urge the authors to include a table in the text explaining which features have been used and why they have been selected. That will complete the revision of the paper, but, in my opinion, it is absolutely mandatory.

Reply: Thank you for your comment. For lunar impact crater age estimation, the 40 morphological features are the generic morphological parameters of craters, and the 38 stratigraphic attributes are related to the crater materials of the five systems, which are listed in the table of attribute data. Information on these attributes can be found at https://github.com/hszhaohs/DeepCraters/tree/master/age_estimation, together with the documentation of each attribute. Meanwhile, the reason to use these attribute data has been described and added in Methods (page 13).

Reviewer: 2

Review by Clark R. Chapman (not anonymous) of the revision of the paper, submitted to Nature Communications, by Chen Yang et al., “Lunar impact crater identification and age estimation with Chang'E data by deep and transfer learning.”

My review of the original paper was encouraging but also highly critical. As I stated, my review dealt with the lunar science in the paper. The machine learning methodologies employed are beyond my technical expertise, so I was not able to evaluate the validity of the methods except by noting that they appeared to identify real craters.

It remains true, in evaluating this revision, that I cannot evaluate the methodology. I have read the other two reviews, apparently by individuals who *are* highly knowledgeable about these methods, and I have read the replies. Clearly both of those reviewers were critical of the methods. But it remains beyond my expertise to evaluate whether or not the authors have correctly addressed those criticisms. So my comments below address *only* the lunar science and the concerns I expressed in my original review.

I am happy to report that the authors have taken my comments very seriously and have made major modifications to their paper that are responsive to my criticisms. Instead of attempting to derive conclusions about lunar science from their data, the focus of the paper has been changed dramatically to *compare* their results with other lunar crater databases. (This was not done at all in the first version of the paper.) The comparisons show good first-order agreement with earlier work (both done visually, or “manually” which is the word the authors use, and by earlier automated algorithms) but they show significant disagreements, as well. Of course, there are equally large disagreements between the other databases. The accumulation of crater data using the traditional visual (“manual”) techniques is extremely time-consuming so such techniques are not likely to be improved. On the other hand, the machine learning approaches are still being rapidly improved and are much more efficient, so they are likely to become the main approach in the future. Thus it is very useful that this paper presents the comparisons with other databases, including comparisons of inferred general ages for the craters.

The original paper attempted some interpretations of the data in terms of lunar science. This material has been removed from this revision, which is good. So the paper has a changed purpose. Instead of attempting to make inferences about lunar evolution it instead offers evidence about how what may become the crater-measurement methodology of the future compares with approaches used during the past 50 years. Such comparisons can motivate further improvements in the methodology. I think that this is a useful contribution to lunar and planetary science. Therefore, on that basis, I would recommend publication of the paper – but with several caveats.

The major caveat is that the other two reviewers must agree that the methodology is technically sound. I cannot make that evaluation myself. Less important, I would note that the English is flawed; this problem is not so severe as to interfere with understanding the paper. But I point it out so that the editor can decide if the English meets the standards of the journal. I would also note that I could offer many “nit-picks” that would further improve the paper. But it would be time-consuming for me to do it, especially since I don’t know the future status of this paper. Because the assignment of ages to the craters is very generalized (to just the five lunar “Systems”) and the chief result of the paper is the comparison of crater numbers and ages with those of other lunar crater studies, I don’t think that my “nit-picks” would be especially important.

To conclude, let me repeat that I am grateful that the authors took my earlier review so seriously

and made such enormous improvements to their paper.

Reply: Thank you very much for your encouragement, suggestions and positive comments on the revised paper. We have addressed the remaining comments of the other reviewers on the machine learning methodologies. Moreover, the English language has been edited and improved with *Springer Nature*.

Reviewer: 3

The authors have done additional testing that aids considerably in evaluating their claims and greatly strengthens the manuscript. The implications for lunar and solar system science have been adequately softened, focusing the manuscript on the technique and resulting dataset, along with empirical trends observed. Clark Chapman is in a better position to evaluate those.

I believe the image data and resulting crater database with estimate age-system classifications, developed using novel deep learning methods, represents a valuable and high-impact contribution worthy of publication in Nature Communications. This is partly due to the authors great decision to make all data publicly available.

I have some remaining concerns on the statistical tests and updated description that should be addressed prior to publication:

1. Several new datasets have been included for testing. This is useful, but makes it unclear what the ground truth is meant to be. The manuscript currently seems to avoid the question and just look at how many craters are recovered from each of 6 different studies in Table 1, but this isn't very satisfactory. There is not much discussion of each dataset, but the combined Povilaitis + Head dataset is a uniform high-fidelity dataset generated by humans with expertise in crater counting / lunar science. If so, it might make sense to combine these two datasets (as done by Povilaitis 2018) and consider that as the ground truth, rather than named craters, which have various biases. Presumably this would be the highest-purity dataset closest to a ground truth, making it the more relevant comparison than the various automated datasets, which may have false positives.

Reply: Thank you for your comments and suggestion. In this paper, the recognized and named craters from the lunar impact craters database of the Lunar and Planetary Institute (LPI) in the analysis of CE-1 images are randomly divided into three separate data sets for training, validation and test in the first stage of the detection approach, and all the recognized craters in CE-2 images are used for testing the second stage of the detection model. There are two reasons for which the recognized and named lunar craters from the International Astronomical Union (IAU) are used for craters identifications: First, it should be acknowledged that the combined Head et al. (2010) and Povilaitis et al. (2018) is a uniform high-fidelity dataset in the scientific research of the Moon. Most of the recognized and named craters are contained within this combined dataset, and a few are distributed in the other crater databases. Second, the purpose of this paper is to use a small number of training data to identify craters effectively and estimate crater age accurately from the same source of data and with the same objective automatic approach. For illustrating the accuracy of identification of detected model, the new identified craters are divided into two set of scales for manual assessment. In the first set, all the identified craters with diameters larger than 100 km (i.e., 218) are involved in the assessment. In the second set, 1% of the other identified craters (i.e., 1171) with diameters between 1 km and 100 km are considered using random selection applied three times and assessing the mean \pm standard deviation (s.d.) error results. The false positive rates of identified craters are listed in Table 1 (page 5).

In order to further verify the reliability of identified craters, the number and percentage matching between currently public lunar crater databases were considered. In the revised paper, the comparison analysis of craters between recognized craters and the currently public lunar crater databases are presented separately; the description of craters comparison and the discussion of each dataset has been reorganized and rewritten in the Results section (pages 4, 5 and 6).

2. It seems like the relevant questions are different at large and small sizes. Currently everything is done at once in Table 1, which makes things unclear. At large sizes: Here presumably the manual crater counts are reliable. So how well does the model do when we know the ground truth?

- The false positive rate is estimated as 1.99%. I do not see how this can be. It is not clear what is meant by the FPR being 'estimated by considering the matchings degree of existing database, and the manual inspection...'

- Looking at Fig 3a: Which database is being compared to in order to label recognized craters?

- Can't one take the database of Head at 20+ km sizes and treat it as complete, and treat all excess craters identified by the model as errors? If we do so then for example at 200-550 km, there are 43 craters, of which 33 are identified. So isn't the precision in that bin 33/43 or about 75% (and about the same for 20+km craters? The manuscript should quote the precision for these larger craters, which is the regime in which it can be accurately determined, or explain why they don't think this is the case. I agree that the precision stops making sense to quote at small sizes where the crater databases become incomplete.

At small sizes: Here's it's unclear what the ground truth is. Table 1 presents what fraction of various automatically detected databases were recovered by the model, but presumably the important question is which of these databases, including the currently presented one, is more reliable at these sizes? There should be some discussion of why different models find widely discrepant numbers of small craters (an order of magnitude more for Robbins 2019). Is it a difference in data or in identification models? Is there any evidence for one dataset being more reliable than another at these sizes? Are there additional data that could help validate/invalidate different models?

Reply: Thank you for your suggestions. In the revised paper, the comparison analysis of craters between recognized craters and the currently public lunar crater databases are presented separately (pages 4, 5 and 6). It is worthy to note that only the recognized and named craters from the IAU are used for test. Fig.3 shows the number of newly identified craters and the recognized craters from the IAU in different diameters scale.

For better comparison with currently public lunar crater databases, the matching percentages of different crater sizes between these databases are supplemented in Table 2 (page 6). The discussion and analysis of the comparison results for craters in large and small sizes have been rewritten in the Results section (page 5).

For the false positive rate, the newly identified craters are then divided into two sets of scales for manual assessment of detection accuracy. In the first set, all the identified craters with diameters larger than 100 km (i.e., 218) are involved in the assessment. In the second set, 1% of the other identified craters (i.e., 1171) with diameters between 1 km and 100 km are considered using random selection applied three times and assessing the mean \pm standard deviation (s.d.) error results. The false positive rates of identified craters are listed in Table 1 (page 5).

3. The false positive rate / precision should only be quoted for the large craters where it can be accurately determined, or quoted separately for small craters and the method for manual counting described. Does the 'manual inspection' mentioned where the FPR is quoted at 1.99% mean that craters of that size were manually identified in a patch by a team member and then compared to the DL detections?

Reply: Thank you for your suggestion. In the revised paper, the false positive rates of large craters

and small craters are quoted, respectively and listed in Table 1. The description and analysis of the false positive rate has been rewritten in the Results section (page 4).

4. Fig 3: Can't tell lines apart in panel b. Reduce number of lines? Reduce number of bins? Only show combined Head and Povilaitis, not both separately in addition. Find a way to show (color, linestyle) that Head + Pov and Robbins are lunar crater databases, and the remainder are automated crater catalogues.

Reply: Thank you for your comment. In the revised paper, Fig 3b (Fig. 4 in the revised paper) has been redrawn taking into account your suggestion (page 6).

5. Fig 6: a) Are the estimations done by taking only the predictions in the 10% holdout test set and then scaling up by a factor of 10? Or are they also showing predictions on craters in the training set? If it's the latter, then the fidelity of the estimates is clearly overemphasized, particularly at large crater sizes.

Reply: For validating effectiveness of the crater classification model, the testing only takes the 10% holdout test set in the first stage with CE1 data. The effectiveness of classification approach in the first stage is tested with CE-1 data on five trials. The best performance model in the first stage is transferred to CE-2 without use any of training data. The craters from 1.26km-50.66km in CE-2 images are used for testing the second stage of classification. The corresponding confusion matrices of the CE-1 and CE-2 data are shown in Fig. 6 (Fig. 7 in the revised paper).

In the lunar impact crater age estimation, the ages of identified craters (i.e., 18,996) larger than 8km in diameter (because small craters degrade at an accelerated with respect to large craters of the same age) are estimated with the best classification model. Fig 7a shows the number of estimated craters and related assigned ages from the Lunar and Planetary Institute (LPI) with five ages in different diameter scales. This is redefined in the revised paper (page 8).

REVIEWER COMMENTS

Reviewer #1 (Remarks to the Author):

The authors replied to my concern about information that was missing, by clarifying where it is possible to find it. I suggest putting the link which is in their reply in the text.

The interest of this work from the point of view of the classification of craters is large. The details of the approach are still a little bit fuzzy (why these features, for instance, and why all of them?), but the software is free and it may be possible to answer these questions by means of the work of other researchers (or the same authors).

Reviewer #3 (Remarks to the Author):

The authors have addressed several of my concerns and improved the manuscript in accordingly.

I still feel like I need to push back on a final point I brought up last round. The FPR is quoted as 2-3%. This is a remarkable statement, appears prominently in the abstract, and thus needs clear substantiation. Perhaps there is a miscommunication, so it would help if the authors responded to this example specifically.

If we look at the 200-550 km bin in Fig 3, it looks like there are ~33 recognized craters, and the model identified 46. So it would seem like the FPR should be > 30%.

If I gather correctly, to estimate the false positives, the authors have instead identified themselves which objects look like craters, and labeled false positives by hand.

Clark Chapman can comment more authoritatively, but presumably there is a community consensus on the crater dataset at these largest sizes. If so, the FPR should be calculated from this community-agreed-upon ground truth. If the community doesn't agree, then it's not clear what a false positive rate really means, and the authors should presumably instead quote a range of FPRs against different authors as a better measure of how confident we can be on the quoted FPRs.

As currently quoted in the abstract, the FPR of 1.82% +/- 0.38% seems both much too low to account for the dispersion across the crater datasets from previous work considered by the authors, and far too precise. It seems (certainly at the small sizes) like this is a problem with no clear ground truth, so the relevant errors to quote on the FPR would be the dispersion across different datasets (different possible ground truths), rather than the much smaller statistical errors on comparisons to manual identifications by the authors.

Clark Chapman would be in a better position to evaluate the level of agreement or disagreement at the large crater sizes.

Reply to the Reviewers' Comments

The authors would like to thank the referees for the valuable comments and suggestions on this paper. We have carefully and thoroughly revised our paper (Numbered NCOMMS-19-38689B) in light of the reviewers' comments. A point-by-point response to the comments is provided below. The major changes have been highlighted by colored text in the revised manuscript.

Reviewer(s) Comments:

Reviewer: 1

1. The authors replied to my concern about information that was missing, by clarifying where it is possible to find it. I suggest putting the link which is in their reply in the text. The interest of this work from the point of view of the classification of craters is large. The details of the approach are still a little bit fuzzy (why these features, for instance, and why all of them?), but the software is free and it may be possible to answer these questions by means of the work of other researchers (or the same authors).

Reply: Thank you for your comment and suggestion. For lunar impact crater age estimation, the stratigraphic coverage relationship is a basic and reliable method. Meanwhile, relative age of craters is estimated according to the degradation and freshness of crater morphology. These determinants of crater age have been described in the Introduction (page 2). Therefore, we selected the generic morphological parameters of craters according to the crater database published by the Lunar and Planetary Institute, and stratigraphic attributes related to the crater materials of the five systems from the 1:5,000,000 Lunar Geologic Renovation (2013 edition).

In the crater age estimation, a dual-channel crater classification model based on the deep neural networks is utilized that takes the raw data (i.e. the craters images and attributes) as input and automatically discovers the representations needed for classification of craters. This is critical property of deep neural architectures. In ref. [1], Yann LeCun, Yoshua Bengio and Geoffrey Hinton reviewed deep learning approaches in *Nature*, 521,436-444 and pointed out the difference between conventional machine-learning techniques and deep learning. The formers are limited in their ability to process natural data in their raw form. They require careful engineering and considerable domain expertise to design a feature extractor that transforms the raw data into a suitable internal representation or feature vector. Unlike conventional techniques, deep learning methods are representation-learning methods with multiple levels of representation, obtained by composing simple but non-linear modules that each transform the representation at one level (starting with the raw input) into a representation at a higher (slightly more abstract) level. Therefore, all these attributes are used as the input to deep neural network to learn representations of data with multiple levels of abstraction in our experiments.

The link of information of 78 attributes and the reason of use all these attribute data have been added and described in Methods (page 13).

[1] LeCun, Y.C., Bengio, Y., & Hinton, G. Deep learning. *Nature*, 521, 436-444 (2015)

Reviewer: 3

The authors have addressed several of my concerns and improved the manuscript in accordingly. I still feel like I need to push back on a final point I brought up last round. The FPR is quoted as 2-3%. This is a remarkable statement, appears prominently in the abstract, and thus needs clear substantiation. Perhaps there is a miscommunication, so it would help if the authors responded to this example specifically.

If we look at the 200-550 km bin in Fig 3, it looks like there are ~33 recognized craters, and the model identified 46. So it would seem like the FPR should be $> 30\%$.

If I gather correctly, to estimate the false positives, the authors have instead identified themselves which objects look like craters, and labeled false positives by hand.

Clark Chapman can comment more authoritatively, but presumably there is a community consensus on the crater dataset at these largest sizes. If so, the FPR should be calculated from this community-agreed-upon ground truth. If the community doesn't agree, then it's not clear what a false positive rate really means, and the authors should presumably instead quote a range of FPRs against different authors as a better measure of how confident we can be on the quoted FPRs.

As currently quoted in the abstract, the FPR of $1.82\% \pm 0.38\%$ seems both much too low to account for the dispersion across the crater datasets from previous work considered by the authors, and far too precise. It seems (certainly at the small sizes) like this is a problem with no clear ground truth, so the relevant errors to quote on the FPR would be the dispersion across different datasets (different possible ground truths), rather than the much smaller statistical errors on comparisons to manual identifications by the authors.

Clark Chapman would be in a better position to evaluate the level of agreement or disagreement at the large crater sizes.

Reply: Thank you for your comments that help us to further improve and refine this part of the paper. In the revised paper, the false positive rates (FPR) of the newly identified craters are recalculated taking into account your suggestion.

Fig.3 has been improved (page 5). In the previous version, all the recognized craters having diameters larger than 1 km and smaller than 500 km located within the study area were considered in the statistics only considering the centre position of crater. In the revised paper, the recognized craters listed in Fig.3 are the ones only completely located within the study area, which are used for crater identification.

The identification of large craters is relatively comprehensive as the number of large craters is much lower than that of the small ones. With regard to the community consensus on the crater datasets, the two existing manual generated databases, i.e., Head et al. (2010) and Robbins (2019) can provide reference for the large craters. However, it should be noted that there are a few craters (e.g. heavily degraded craters) that cannot be identified by the manual detection. For the small craters, Povilaitis et al. (2018) and Robbins (2019) could provide only partly reference due to the large number of small ones. Therefore, the newly identified craters, i.e., 11145 craters which are not included in the recognized craters are divided into two sets of scales for manual assessment of detection accuracy. In the first set, all the newly identified craters with diameters larger than 100 km (i.e., 166) are involved in the assessment. In the second set, 10% of the other identified craters in different regions and scales (i.e., 10979) with diameters between 1 km and 100 km are considered using a statistical sampling by random selection. These craters are assessed by matching with three existing manual databases and independently inspected by four scientists

from Key Laboratory of Lunar and Deep Space Exploration, Chinese Academy of Sciences, simultaneously. The selected newly identified craters are projected onto CE-2 7m images and the mean \pm standard deviation (s.d.) error results are provided. Meanwhile, the FPR of 361 new craters in the only DL-based model i.e., Silburt et al. (2019) is quoted, which is implemented by manual inspection also with four scientists from their research group and averaging the results. The description of FPR of newly identified craters has been reorganized and rewritten in the Results section (page 5). The false positive rates of newly identified craters are listed in Table 2 (page 5).

REVIEWERS' COMMENTS

Reviewer #3 (Remarks to the Author):

I appreciate the authors' efforts to address my concerns and incorporate them into the paper. While I do not feel like their method of estimating the FPR captures the uncertainties in the underlying ground truth, the paper clearly explains their methodology for calculating the FPR.

I am happy to accept to accept the paper, and congratulate the authors for both the impressive technical achievement and for openly sharing it with the scientific community.

Reply to the Reviewers' Comments

The authors would like to thank the referees for the valuable comments and suggestions on this paper. We have carefully and thoroughly revised our paper (Numbered NCOMMS-19-38689C) in light of the reviewers' comments.

Reviewer(s) Comments:

Reviewer: 3

I appreciate the authors' efforts to address my concerns and incorporate them into the paper. While I do not feel like their method of estimating the FPR captures the uncertainties in the underlying ground truth, the paper clearly explains their methodology for calculating the FPR.

I am happy to accept to accept the paper, and congratulate the authors for both the impressive technical achievement and for openly sharing it with the scientific community.

Reply: Thank you very much for your encouragement, suggestions and comments on the revised paper. About the uncertainties in the underlying ground truth, the subjectivity of manual detection with different types of data has resulted in disagreement in existing lunar crater databases. For the FPR, we estimate craters in different regions and scales by random selection to mitigate the uncertainties. Thank you again for your comments that helped us to improve and refine the paper. On behalf of my co-authors, we would like to express our great appreciation to your review.